# A commensal protozoan attenuates *Clostridioides difficile* pathogenesis in mice via arginine-ornithine metabolism and host intestinal immune response

Huan Yang[1,7], Xiaoxiao Wu[1,7], Xiao Li[1,7], Wanqing Zang[1], Zhou Zhou[1], Yuan Zhou [1], Wenwen Cui[2], Yanbo Kou [3], Liang Wang [4], Ankang Hu[5], Lianlian Wu[5], Zhinan Yin[6], Quangang Chen[5], Ying Chen [1], Zhutao Huang[5], Yugang Wang [3] ✉ & Bing Gu [1,4] ✉

Antibiotic-induced dysbiosis is a major risk factor for *Clostridioides difficile* infection (CDI), and fecal microbiota transplantation (FMT) is recommended for treating CDI. However, the underlying mechanisms remain unclear. Here, we show that *Tritrichomonas musculis* (*T.mu*), an integral member of the mouse gut commensal microbiota, reduces CDI-induced intestinal damage by inhibiting neutrophil recruitment and IL-1β secretion, while promoting Th1 cell differentiation and IFN-γ secretion, which in turn enhances goblet cell production and mucin secretion to protect the intestinal mucosa. *T.mu* can actively metabolize arginine, not only influencing the host's arginine-ornithine metabolic pathway, but also shaping the metabolic environment for the microbial community in the host's intestinal lumen. This leads to a relatively low ornithine state in the intestinal lumen in *C. difficile*-infected mice. These changes modulate *C. difficile*'s virulence and the host intestinal immune response, and thus collectively alleviating CDI. These findings strongly suggest interactions between an intestinal commensal eukaryote, a pathogenic bacterium, and the host immune system via inter-related arginine-ornithine metabolism in the regulation of pathogenesis and provide further insights for treating CDI.

*Clostridioides difficile* (*C. difficile*) is a Gram-positive, spore-forming, anaerobic bacillus. *C. difficile* spore, conferring resistance to adverse environment and allowing persistence for several months, is the infectious agent of antibiotic-associated diarrhea[1,2]. Treatment with broad-spectrum antibiotics disrupts the naturally diverse gut microbiota, enabling the growth of *C. difficile*, including strains that can produce toxins that disrupt the intestinal barrier[3]. The worldwide spread of *C. difficile* infection (CDI) imposes a huge economic burden

[1]Xuzhou Key Laboratory of Laboratory Diagnostics, School of Medical Technology, Xuzhou Medical University, Xuzhou, Jiangsu, China. [2]Xuzhou Center for Disease Control and Prevention, Xuzhou, Jiangsu, China. [3]Laboratory of Infection and Immunity, Jiangsu Key Laboratory of Immunity and Metabolism, Department of Pathogenic Biology and Immunology, Xuzhou Medical University, Xuzhou, Jiangsu, China. [4]Department of Clinical Laboratory Medicine, Guangdong Provincial People's Hospital, Southern Medical University, Guangzhou, Guangdong, China. [5]Center of Animal Laboratory, Xuzhou Medical University, Xuzhou, Jiangsu, China. [6]The Biomedical Translational Research Institute, Health Science Center (School of Medicine), Jinan University, Guangzhou, Guangdong, China. [7]These authors contributed equally: Huan Yang, Xiaoxiao Wu, Xiao Li. ✉e-mail: wangyg@xzhmu.edu.cn; gb20031129@163.com

on human societies[4,5]. Antibiotics are the main therapy for CDI. However, the acquisition of resistance to commonly used antibiotics facilitates the epidemic spread of hypervirulent *C. difficile* strains[6]. Antibiotics such as metronidazole, vancomycin, and fidaxomicin are still in use for treating CDI, but there are challenging threats to face, such as adverse drug reactions, CDI recurrence, and multidrug-resistant strains emergence[7,8]. Therefore, increasing attention has been paid to non-antibiotic treatment of CDI in recent years.

Emerging non-antibiotic therapies include fecal microbial transplantation (FMT)[9,10], probiotics[11,12], and dietary interventions[13–15]. Different from the bactericidal effect of antibiotics, these therapies are more likely to regulate gut microbiota to inhibit the germination and growth of *C. difficile* spores and enhance host immunity to reduce the damage caused by *C. difficile*[16,17]. FMT, widely used to treat recurrent CDI with clinical success rates more than 90%[9], is a therapy aimed at restoring the altered intestinal microbiota by administrating fecal microorganisms from a healthy donor into the intestine of a patient. Most research on the role and mechanism of FMT have focused on the bacterial microbiota of the gut, while the role of other microorganisms, such as fungi, viruses, archaea, and protozoa, has not received sufficient attention.

In normal conditions, gut microbes help with the digestion and absorption of nutrients, prevent colonization of pathogens, and promote host immunity[18]. Most studies on the probiotic effects of intestinal microorganisms come from prokaryotes such as bacteria, while eukaryotes such as fungi, worms, and protozoa are often found to have adverse effects on the host, such as *Entamoeba histolytica*[19,20], *Giardia lamblia*[21] and *Cryptosporidium*[22]. While Chudnovskiy and colleagues discovered a commensal protozoan *Tritrichomonas musculis* (*T.mu*) in healthy mice, which activates host intestinal epithelial inflammasome, induces IL-18 release, and promotes dendritic cell-driven Th1 and Th17 immunity[23]. *T.mu* colonization was confirmed to affect the intestinal microbiota[24]; however, Chudnovskiy and colleagues showed that *T.mu* was sufficient to induce Th1 and Th17 cells in the germ-free mice, indicating that other components of the gut microbiota may not be necessary for the immune modulative effects of *T.mu*[23]. In addition, *Tritrichomonas spp*. can produce a great amount of succinate by their intracellular special organelle named hydrogenosome, and succinate can not only activate innate lymphoid-like cell (ILC)−2 via a tuft cell-IL25-dependent pathway but also induce intestinal tuft cell and Paneth cell expansion via a type 2 immunity-dependent manner[25–28].

Notably, anaerobic trichomonads obtain energy mainly from carbohydrate and amino acid metabolism[29]. Studies indicate that trichomonads require arginine for growth and the arginine dihydrolase pathway has the ability to provide ATP to the protozoal cells[30,31]. Arginine is known as a host immune booster, but the mechanism remains poorly defined[32]. The metabolic activities of trichomonads have great impact on host metabolism. Recently research has demonstrated that *T.mu* colonization can influence host glucose metabolism by facilitating free choline generation[33]. However, it is unclear whether *T.mu* can also influence host amino acid metabolism.

Here, we introduce *T.mu* into the CDI model, aiming to explore how a non-bacterial commensal member of the microbiota contribute to the CDI pathogenesis, which may be helpful to clarify the therapeutic principle of FMT and develop biologic agents for the treatment of CDI. Our results strongly suggest an interaction of *T.mu*, *C. difficile*, and host immunity possibly via an amino acid-dependent metabolic pathway to affect the host's susceptibility to CDI.

## Results

### *T.mu* colonization in the gut alleviates CDI-induced inflammation and intestinal damage

As previously described, we had successfully isolated and identified a mouse commensal protozoan that is genetically identical to *T.mu* from our in-house mouse colony[24]. Scanning electron microscopy showed that *T.mu* was pear-shaped with three anterior flagella and one posterior flagella (Supplementary Fig. 1A).

To assess the influence of *T.mu* on the mice health, WT B6 mice were administrated with purified *T.mu* once every other day for a total of four times via oral gavage, and the mouse body weight was then monitored daily. As we need to use antibiotics to build the mouse CDI model, the influence of the antibiotics treatment on *T.mu*'s colonization in the gut was also monitored. We found that the body weight changes were similar between *T.mu*-treated and non-treated mice (Supplementary Fig. 1B), and the mice had no diarrhea and hunchback post *T.mu* inoculation. In addition, more *T.mu* was colonized in the antibiotic-treated mice (Supplementary Fig. 1C). These data suggest that *T.mu* does not have an obvious influence on the health of the mice, and the antibiotics regimen designed for our experiment does not inhibit *T.mu* colonization.

Next, to assess the possible effects of *T.mu* on CDI, we established a *T.mu* and *C. difficile* co-colonization model as depicted in Fig. 1A. We found that *T.mu* colonization significantly alleviated CDI-induced weight loss (Fig. 1B), and greatly reduced disease activity index (DAI) (Fig. 1C). After 48 h post infection, compared with mice in the CDI group, the mice in the CDI + *T.mu* group exhibited relatively less intestinal epithelial damage (Fig. 1D), longer colon length (Fig. 1E), lower histological score (Fig. 1F), and more goblet cells (Fig. 1G, H). These results indicate that *T.mu* can significantly relieve CDI-induced intestinal inflammation.

CDI did not influence *T.mu* colonization, as the *T.mu* counts were no difference in the fecal and cecal content in the *T.mu*-colonized mice with or without *C. difficile* (Supplementary Fig. 1D). Vice versa, *T.mu* colonization also did not have obvious impact on *C. difficile* colonization in the gut, as comparable amount of *C. difficile* vegetative and spore biomass was recovered in the faeces the cecal contents and cecum between the CDI group and the CDI + *T.mu* group (Fig. 1I–L). Notably, the toxin levels produced by *C. difficile*, as represented by TcdB titers detected in the fecal content at 12 h and 24 h post infection were lower in the CDI + *T.mu* group compared with CDI group (Fig. 1M). In conclusion, *T.mu* colonization does not influence *C. difficile* colonization capacity, instead it attenuates its virulence, therefore alleviating CDI-induced inflammation and intestinal damage.

### *T.mu* colonization in GF mice can directly protect against *C. difficile* infection

To investigate whether *T.mu* colonization in the gut influences the gut microbiota especially during CDI, we performed bacterial 16 s rRNA gene (V3–V4 region) sequencing of cecal samples isolated from *C. difficile*-infected or non-infected mice with or without *T.mu* inoculation. Linear discriminant analysis effect size (LEfSe) was performed to identify differential abundance in bacterial compositions. As shown in the Supplementary Fig. 2A, *T.mu* colonization greatly changes the bacterial microbiota landscape. The dominant differential bacterial families in the normal control (NC) group were *Lachnospiraceae*, *Erysipelatoclostridiaceae*, and *Bifidobacteriaceae*, while the dominant differential bacterial families were *Bacteroidaceae*, *Akkermansiaceae*, and *Ruminococcaceae* in the NC + *T.mu* group. After CDI, the dominant one in the CDI group changed to *Enterobacteriaceae*, while in CDI + *T.mu* group *Erysipelotrichaceae*, *Oscillospiraceae*, *Peptostreptococcaceae*, and *Prevotellaceae* became the dominant differential bacteria (Supplementary Fig. 2B). Notably, the relative abundance of *Enterococcaceae* was decreased in CDI + *T.mu* group compared to CDI group (Supplementary Fig. 2C, D), and *Enterococci* in this bacterial family are known to be able to enhance *C. difficile* pathogenesis[34]. Thus, *T.mu* colonization or not indeed influences the microbial community that *C. difficile* might encounter during its infection.

To investigate whether *T.mu* can directly mediate protection against CDI without a need of help from other microbiota members,

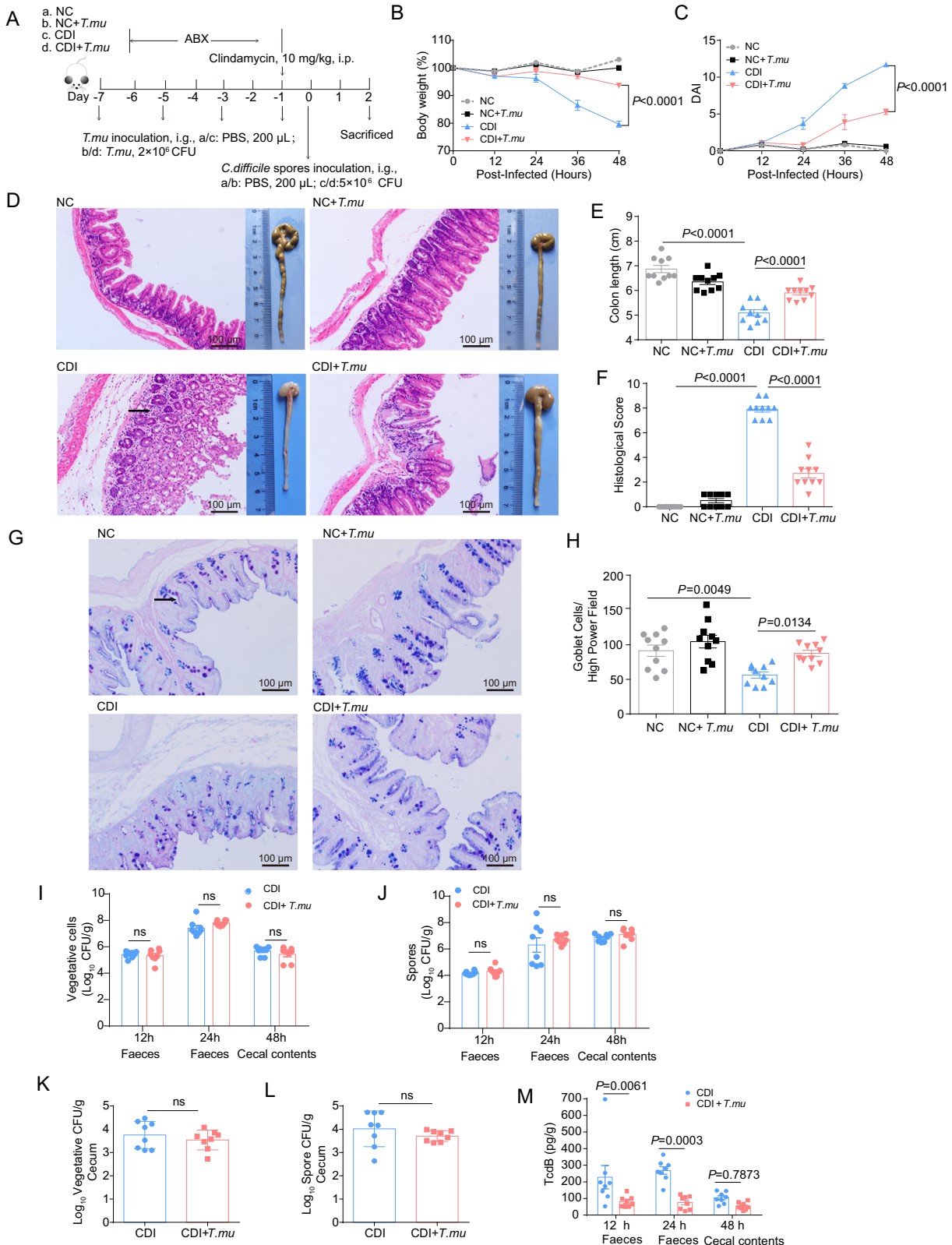

we first treated *T.mu*-colonized mice with 3 mg/mL metronidazole (MNZ) for a week to remove *T.mu* from the mice (Supplementary Fig. 2E). CDI-induced-pathology was observed to a similar extent in both the *T.mu*-depleted "CDI + *T.mu* (MNZ)" group and the CDI group (Supplementary Fig. 2F–I), suggesting of a strong association between the need of the presence of *T.mu* and the CDI protection.

Next, we colonized germ-free (GF) mice with purified *T.mu* and one week later probed its effect on *C. difficile* infection. We found that the body weight changes were similar between *T.mu* mono-colonized and non-treated mice (Fig. 2A). GF mice without *T.mu* colonization, the CDI group developed symptoms, such as weight loss and diarrhea, at 36 h post *C. difficile* challenge (Fig. 2B, C), and all the mice died within

**Fig. 1 | *T.mu* colonization in the gut alleviates CDI-induced inflammation and intestinal damage.** On days −7 to −1, mice in the NC + *T.mu* and CDI + *T.mu* groups were treated with purified *T.mu*, respectively. On day 0, each mouse in CDI group and CDI + *T.mu* group was given *C. difficile* spores. Cecum and colon of mice were collected at day 2 post infection. NC normal mice without *T.mu*. NC + *T.mu*: normal mice treat with *T.mu*. CDI: mice treat with *C. difficile*. CDI + *T.mu*: mice treat with *C. difficile* and *T.mu*. **A** Schematic of the experimental design. **B** Body weight changes post CDI (*n* = 10 per group). **C** Disease activity index (DAI) (*n* = 10 per group). **D** Macroscopic photos of colon and representative images of H&E-stained cecum tissue sections from the indicated group of mice. Scale bar: 100 μm. Arrow indicates infiltration of inflammatory cells. **E** Measurement of the colon length (*n* = 10 per group). **F** Histological score of cecal tissues collected from the indicated mice (*n* = 10 per group). **G** Representative PAS staining of cecal tissue sections from the indicated mice. Scale bar: 100 μm. Arrow indicates PAS-stained-positive goblet cells. **H** Quantitation of the goblet cell numbers in the cecal tissues (*n* = 10 per group). **I, J** The numbers of vegetative cells (**I**) and spores (**J**) of *C. difficile* in the faeces 12 or 24 h post infection, and the numbers of vegetative cells and spores in cecal contents 48 h post infection (*n* = 8 per group). **K, L** The numbers of vegetative cells (**K**) and spores (**L**) of *C. difficile* in the cecum 48 h post infection (*n* = 8 per group). **M** The levels of TcdB in the faeces 12 or 24 h post infection, and the levels of TcdB in the cecal contents 48 h post infection (*n* = 8 per group). Experiments were repeated independently three times. Data are the mean ± SEM. Statistical significance was determined by two-way ANOVA (**B, C, I, J, M**), one-way ANOVA (**E, F, H**) or two-sided Student's *t* test (**K, L**). ns no significance. Source data are provided as a Source Data file.

48 h (Fig. 2D). In contrast, GF mice that colonized with *T.mu*, the CDI + *T.mu* group developed very little symptoms and all survived during *C. difficile* infection (monitored for 10 days) (Fig. 2B–D). Compared with control mice, *T.mu*-colonized GF mice had similar burdens of *C. difficile* vegetative (Fig. 2E) and spore biomass (Fig. 2F) but had significantly lower titers of TcdB in the fecal contents, especially at 24 h and 36 h after infection (Fig. 2G). Furthermore, *T.mu*-colonized GF mice had relatively intact intestinal epithelium, longer colon and much lower histological score post infection (Fig. 2H, I). The number of goblet cells in the colon was also increased in *T.mu*-colonized GF mice (Fig. 2J, K). In addition, the expression levels of inflammatory cytokines CXCL1 and IL-1β were reduced, while the IFN-γ expression levels were increased (Fig. 2L–O). Therefore, our results demonstrate that *T.mu* colonization can directly mediate protection against CDI independent of the altered microbial community caused by *T.mu*.

## *T.mu* colonization reduces neutrophil recruitment to the *C. difficile* infection sites

Neutrophils have been shown to be extremely important in the early host response to CDI[35]. In our *T.mu*-*C. difficile* co-colonization model, *C. difficile* infection significantly promoted the neutrophil recruitment into the colon lamina propria compared to normal mice, while *T.mu* colonization alone did not affect the neutrophil recruitment, however, *T.mu* surprisedly reduced the percentage and number of neutrophils accumulated in the colon after CDI (Fig. 3A). The neutrophils accumulated in the colon after CDI are likely mobilized from the bone marrow, as the ratio and number of neutrophils in the bone marrow were significantly dropped after infection (Fig. 3B). Our data suggested that *T.mu* colonization significantly reduced CDI-induced neutrophil mobilization from the bone marrow and the recruitment of neutrophils from the blood into the colonic sites of infection. In accordance with this, the expression of cytokines CXCL1, IL-36γ, IL-1β, and IL-6, which are thought to be critical in the recruitment of neutrophils to the infection sites and promoting inflammation, were significantly reduced in the CDI + *T.mu* group compared to the CDI group (Fig. 3D–K). Together, our data suggest that *T.mu* colonization can temper vigorous recruitment of neutrophils during infection, possibly preventing from overactive immune activation-induced tissue damage.

Next, we examined whether *T.mu* can directly influence neutrophil function. We set up an in vitro *T.mu*-*C. difficile*-neutrophils co-cultured system, *C. difficile* induced plentiful IL-1β secretion from neutrophils, while *T.mu* inhibited *C. difficile*-induced IL-1β secretion with several different ratios of *C. difficile*/*T.mu* (Fig. 3L). These results suggest that *T.mu* may relieve *C. difficile*-induced enteritis by fine-tuning, decreasing the recruitment of neutrophils and the secretion of neutrophils-related cytokines.

## *T.mu* protects intestinal mucosa by increasing intestinal IFN-γ

Studies have shown that *Tritrichomonas* can promote the induction of T helper (Th)1 and Th17 responses[23,36]. We therefore investigated the influence of *T.mu* on the phenotypical traits of T cells during CDI. The gating strategy for analysis of Th1 cells from mice colonic lamina propria lymphocytes (LPLs) was shown in Supplementary Fig. 3B. After *C. difficile* infection, *T.mu* significantly increased the percentage and numbers of Th1 cells but not Th17 cells in the colon (Fig. 4A, B). Consistent with the dominant Th1 cell induction phenotype, we observed a higher level of *Ifng* gene and IFN-γ protein expression in the colon of CDI + *T.mu* group compared to the CDI group (Fig. 4C, D), while *T.mu* did not influence the expression of IL-17A in the colon with or without CDI (Fig. 4E).

In addition to Th1 cells that can produce IFN-γ, other immune cells, such as innate lymphoid cells (ILCs), cytotoxic T (Tc)1 cells, and γδ T cells can also secrete IFN-γ. Therefore, we also investigated whether *T.mu* colonization can alter these immune cells in the lamina propria. Our results showed that the *T.mu* had no significant impact on the numbers and percentages of ILCs (Supplementary Fig. 3C, D), macrophages (Supplementary Fig. 3E), γδ T cells (Supplementary Fig. 3F), and Tc1 cells (Supplementary Fig. 3G) upon CDI. These results suggest that *T.mu* colonization mainly promotes the secretion of IFN-γ by Th1 cells to protect against CDI.

Since parasites often enhance type II immunity in the host gut, we also tested type 2 cytokines in *T.mu*-colonized mice. The results indicated that *T.mu* did not significantly influence the expression of intestinal cytokines IL-4 and IL-13 (Supplementary Fig. 3H, I). Therefore, *T.mu* may mainly protect against CDI by promoting Th1 type response.

To investigate whether IFN-γ play an important role in the protection against CDI, 10 μg recombinant IFN-γ was intraperitoneally injected into mice at 2 h p.i. and 24 h p.i. We found that IFN-γ treatment greatly improved *C. difficile*-induced body weight loss (Supplementary Fig. 4A), reduced the DAI (Supplementary Fig. 4B), and alleviated the intestinal epithelium damage with longer colon length and lower histological score (Supplementary Fig. 4C–E), as well as the *T.mu* treatment. Moreover, IFN-γ treatment alleviated *C. difficile*-induced goblet cell and MUC2 reduction (Supplementary Fig. 4F, G). Together, these results demonstrate that IFN-γ plays an important role in protecting against CDI.

To further explore whether *T.mu* conduct the protective effect on CDI through IFN-γ, we colonized *T.mu* to WT mice and *Ifngr*−/− mice, then infected the mice with *C. difficile*. The protective effect of *T.mu* colonization on CDI was observed in the CDI + *T.mu* group compared to the CDI group in WT mice, however, *T.mu* could not relieve the body weight loss and clinical symptoms caused by *C. difficile* in *Ifngr*−/− mice (Fig. 4F, G). Compared to the WT mice that colonized with *T.mu* and infected with *C. difficile* (labeled as the CDI + *T.mu* group), the *Ifngr*−/− mice that colonized with *T.mu* and infected with *C. difficile* (labeled as the CT+Ifngr−/−group) had similar degrees of colonic atrophy, colon shortening, inflammatory cell infiltration (Fig. 4H–J), and goblet cell and MUC2 decreasing (Fig. 4K, L), suggesting that *T.mu*'s effective protection from CDI requires IFN-γ signaling. Moreover, there were a slightly increased

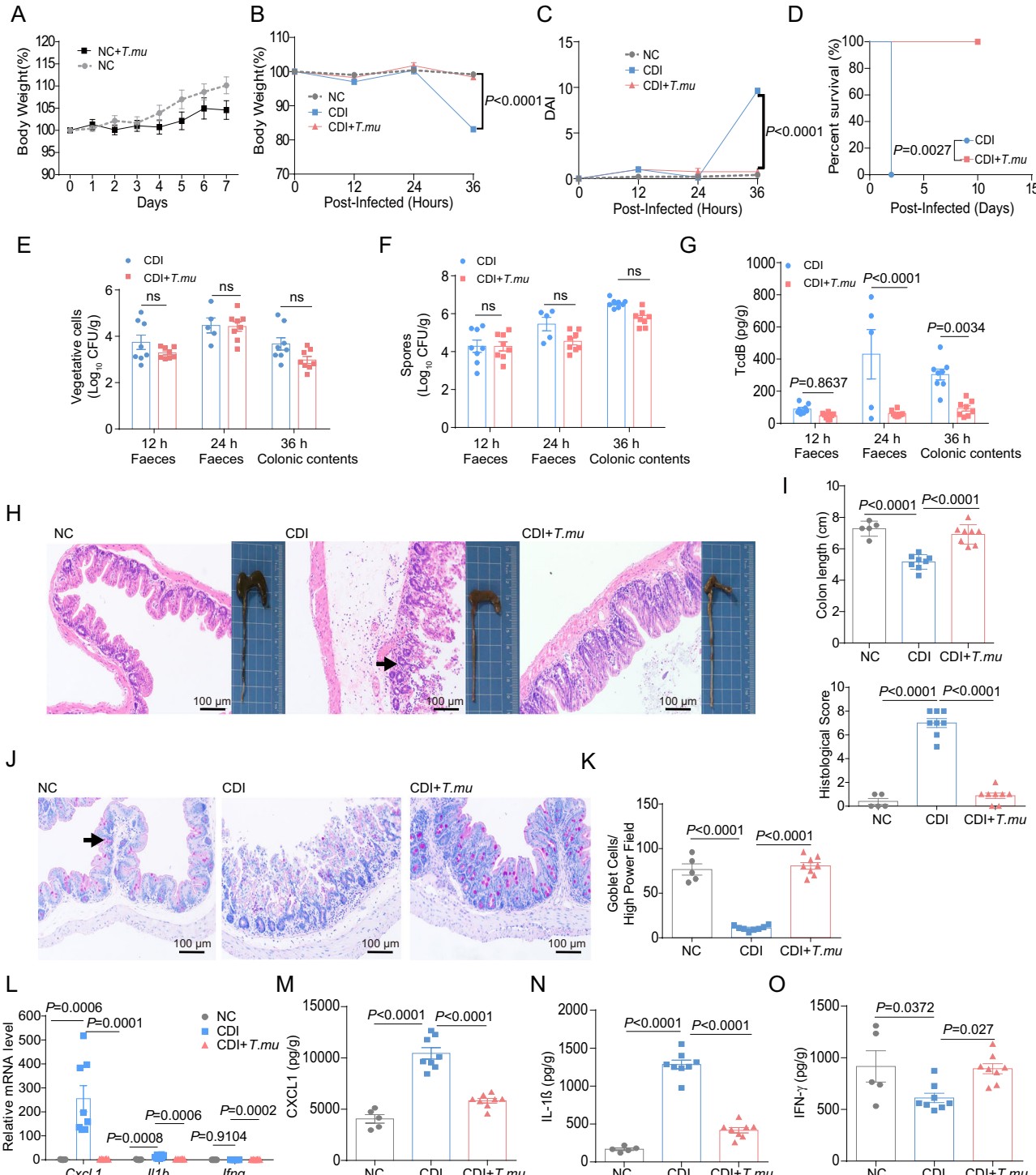

**Fig. 2 | *T.mu* colonization in germ-free (GF) mice can directly protect against lethal *C. difficile* infection.** GF B6 mice were divided into 3 groups: normal control group (NC), *C. difficile*-infected group (CDI) and *T.mu* plus *C. difficile* group (CDI + *T.mu*). On day −7, mice in the CDI + *T.mu* group were gavaged orally with ex vivo purified *T.mu*. On day 0, each mouse in the CDI group and CDI + *T.mu* group was given *C. difficile* spores. Mice were sacrificed at 36 h after infection. **A** Body weight changes after *T.mu* colonization (*n* = 5 in the NC group, *n* = 8 in NC + *T.mu* group). **B** Body weight changes post CDI (*n* = 5 in the NC group, *n* = 8 in the CDI and CDI + *T.mu* group). **C** Disease activity index (DAI) (*n* = 5 in the NC group, *n* = 8 in the CDI and CDI + *T.mu* group). **D** Survival curve (*n* = 5 per group). **E**–**G** The numbers of *C. difficile* vegetative cells (**E**), spores (**F**), and the levels of TcdB (**G**) in the faeces (at 12 or 24 h post infection) and the colonic contents (at 36 h post infection) (with the exception of *n* = 5 in the CDI group 24 h after infection, *n* = 8 for other groups). **H** Macroscopic photos of colon and representative microscopic images of HE-stained cecal tissue sections. Scale bar: 100 μm. Arrow indicates infiltration of inflammatory cells. **I** Measurement of the colon length and quantitation of cecal tissue histological score (*n* = 5 in the NC group, *n* = 8 in the CDI and CDI + *T.mu* group). **J** Representative PAS staining of cecal tissue sections from the indicated mice. Scale bar: 100 μm. Arrow indicates goblet cells. **K** Quantitation of the goblet cell numbers in the cecal tissues (*n* = 5 in the NC group, *n* = 8 in the CDI and CDI + *T.mu* group). **L** The *Cxcl1*, *Il1b*, and *Ifng* levels in the colon were determined using qRT-PCR (*n* = 5 in the NC group, *n* = 8 in the CDI and CDI + *T.mu* group). **M**–**O** The levels of (**M**) CXCL1, (**N**) IL−1β, and **O** IFN-γ in the cecum were determined by ELISA (*n* = 5 in the NC group, *n* = 8 in the CDI and CDI + *T.mu* group). Experiments were repeated independently two times. Data are mean ± SEM. Statistical significance was determined by two-way ANOVA (**A**–**G**), one-way ANOVA (**I**, **K**, **L**–**O**). ns no significance. Source data are provided as a Source Data file.

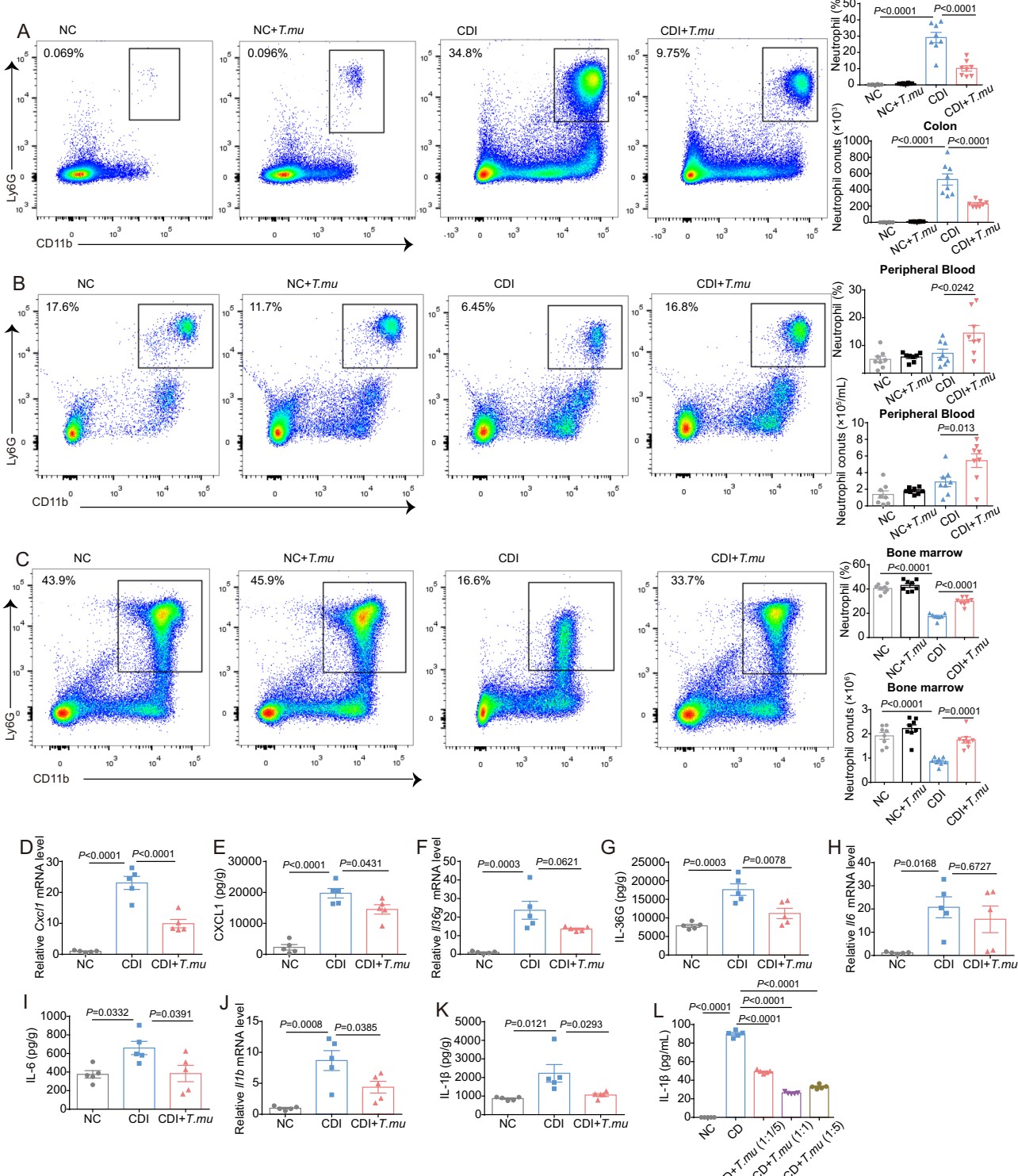

**Fig. 3 | *T.mu* colonization reduces neutrophil recruitment to *C. difficile* infection sites. A**–**L** The colon, cecum, peripheral blood, and bone borrow from the indicated mice were collected at day 2 post *C. difficile* infection. **A**–**C** Representative flow cytometry graphs and the neutrophils percentages and numbers in (**A**) the colonic lamina propria, **B** peripheral blood, and **C** bone borrow (*n* = 8 per group). The relative expression levels of **D** *Cxcl1*, **F** *Il36g*, **H** *Il6*, and **J** *Il1b* in the colon were determined by qRT-PCR. The concentrations of **E** CXCL1, **G** IL-36G,

**I** IL-6, and **K** IL-1β in the cecum were determined by ELISA (*n* = 5 per group). **L** Different ratios of *C. difficile* vs. *T.mu* (*C. difficile*: *T.mu* = 1:0, 1:0.2, 1:1, or 1:5, respectively) were mixed with neutrophils in vitro (*n* = 6 per group). Experiments were repeated independently two times. Data are the mean ± SEM. Statistical significance was determined by one-way ANOVA (**A**–**L**). Source data are provided as a Source Data file.

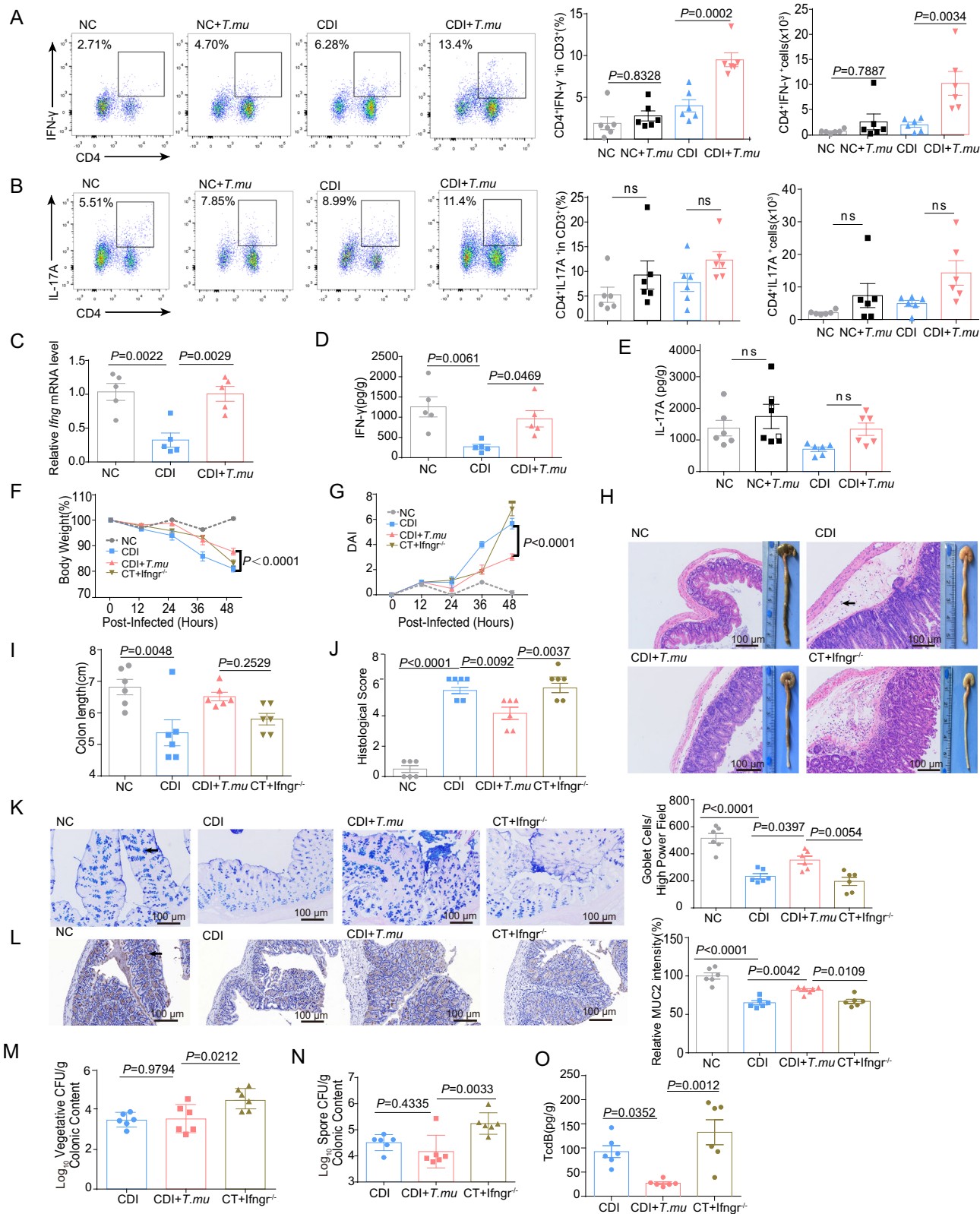

levels of *C. difficile* spore biomass, and a trend of increasing TcdB levels in the feces of *Ifngr*⁻/⁻ mice in the CT+Ifngr⁻/⁻ group, compared to the WT mice in the CDI + *T.mu* group (Fig. 4M−O). Collectively, these observations suggest that *T.mu* protects the mucosa from CDI-induced damage possibly by an *Ifng*-dependent pathway that preserving the stability of goblet cells to produce mucus during infection.

## *T.mu* colonization influence gut microbial community and host arginine/ornithine metabolism

Studies have shown that *Tritrichomonas spp.* can utilize amino acids to produce energy[30,31]. We found that *T.mu* that cultured in a rich media– Brain Heart Infusion (BHI) broth in vitro for 24 h consumed great amount of arginine and released citrulline and ornithine into the media (Supplementary Fig. 5A−C), suggesting that *T.mu* may be able to

**Fig. 4 | T.mu protects intestinal mucosa by increasing intestinal IFN-γ. A–E** The colon and cecum from the indicated mice were collected at day 2 post *C. difficile* infection. Colonic LP cells were incubated in the presence of brefeldin A, ionomycin, and PMA for 5 h and then stained with anti-CD45, CD3, CD4, IL-17, and IFN-γ. IFN-γ, and IL-17 production by CD4 + T cells was analyzed using flow cytometry. Representative flow cytometry graphs and the percentages and numbers of **A** Th1 and **B** Th17 cells in colonic lamina propria (*n* = 6 per group). **C** The *Ifng* mRNA levels in the colon were determined by qRT-PCR (*n* = 5 per group). The levels of **D** IFN-γ, and **E** IL-17A in the cecum were determined by ELISA (*n* = 6 per group). **F–O** The *Ifngr*$^{-/-}$ mice were gavaged with purified *T.mu* every other day for a total of one week, and then the mice were infected with *C. difficile* (CT+Ifngr$^{-/-}$ group). WT B6 mice were used as controls (CDI + *T.mu*). The cecum and colon from the indicated mice were collected at day 2 post infection. **F** Body weight changes (*n* = 6 per group). **G** DAI (*n* = 6 per group). **H** Macroscopic findings of colon and representative HE-staining images of cecal tissue sections. Scale bar: 100 μm. Arrow indicates infiltration of inflammatory cells. **I** Measurement of the colon length (*n* = 6 per group). **J** Histological score for (*n* = 6 per group). **K** Quantitation of the colonic goblet cells and representative goblet cell staining images (*n* = 6 per group). Scale bar: 100 μm. Arrow indicates goblet cells. **L** Relative expression intensity of MUC2 and representative MUC2 histochemical staining images (*n* = 6 per group). Scale bar: 100 μm. Arrow indicates MUC2-positive stain. The number of *C. difficile* (**M**) vegetative cells, **N** spores, and the levels of (**O**) TcdB in the cecal contents at 48 h post infection (*n* = 6 per group). Experiments were repeated independently two times. Data are the mean ± SEM. Statistical significance was determined by two-way ANOVA (**F, G**) or one-way ANOVA (**A–E, I–O**). ns no significance. Source data are provided as a Source Data file.

actively metabolize arginine, possibly by arginine dihydrolase pathway for energy use. Meanwhile, we found that in the same culture condition *C. difficile* consumed little arginine and produced an insignificant amount of ornithine and citrulline (Supplementary Fig. 5A–C).

Next, we asked whether *T.mu* colonization can influence overall gut microbiome metabolomes. We therefore conducted untargeted metabolomic profiling of cecal content samples collected from *C. difficile*-infected mice with or without *T.mu* colonization. We found that many of the differentially enriched metabolites in the CDI group were amino acid metabolic intermediates (Fig. 5A), and within these metabolites, many of them were intermediates participating in the arginine biosynthesis pathway (Fig. 5B). Compared with the CDI group, the relative levels of ornithine in the cecal contents were decreased, while the citrulline levels were instead increased in the CDI + *T.mu* group (Fig. 5C), suggesting a role of *T.mu* colonization in influencing certain amino acids richness (e.g., arginine/ornithine, etc.) in the gut lumen, a critical space shared by the gut microbial community.

We then performed targeted metabolomic quantitation of several selected metabolic intermediates involved in arginine/ornithine metabolism related to both *T.mu* and *C. difficile* (Fig. 5D, E). Our data indicated that *T.mu* colonization in normal control WT mice without *C. difficile* infection resulted in a reduction of arginine and increasing of ornithine, citrulline, putrescine, and 5-aminovalerate in the cecal contents (Fig. 5F, G). CDI dramatically reduced the levels of arginine and 5-aminovalerate and increased the levels of ornithine in the cecal contents, compared to the uninfected normal mice control (Fig. 5F, G). In the presence of *T.mu*, *C. difficile*-induced increase of ornithine in the cecal contents were tempered (Fig. 5F), while the level of putrescine and alanine was increased, compared to the CDI group (Fig. 5G).

*T.mu* colonization also changed the levels of arginine/ornithine metabolic intermediates in the colon tissues in the presence or absence of *C. difficile* infection (Fig. 5H). The host's arginine metabolizing enzymes include arginase 1 (ARG1), inducible nitric oxide synthase (iNOS), arginine succinate synthetase 1 (ASS1), ornithine transcarbamylase (OTC), and ornithine decarboxylase (ODC) (Fig. 5J–L). Notably, compared to the CDI group, the host's metabolic enzyme genes *Nos2* (encoding for iNOS), *Ass1*, and *Otc* in the colon tissues were upregulated, and *Arg1* was downregulated in the CDI + *T.mu* group (Fig. 5J). Immunofluorescence staining indicated that, compared to the CDI group, the number of iNOS-positive cells and the number of ASS1-positive cells in the colon were all increased, while the number of ARG1-positive cells was reduced in the CDI + *T.mu* group (Fig. 5L–N). These results together suggest that *T.mu* colonization can greatly influence the gut microbial community metabolomes and the host gut tissue metabolic activities, especially the cross-kingdom inter-related metabolic pathways involved in arginine–ornithine metabolism.

Next, we asked whether the influence of *T.mu* on CDI-induced arginine/ornithine metabolic perturbation is dependent on the gut microbiota. To this end, we colonized GF mice with *T.mu* and

challenged them with *C. difficile* one week later, and then collected the cecal contents, colon tissues, and sera 36 h post infection. Targeted metabolomic quantitation indicated that *T.mu* colonization significantly reduced CDI-induced arginine and ornithine upregulation in the cecal contents, while increased the levels of putrescine, alanine, and 5-aminovalerate compared to CDI mice (Supplementary Fig. 6A, B). In addition, compared to the CDI group, the host's colonic *Nos2* and *Otc* expression levels were upregulated, and *Arg1* expression was downregulated in the CDI + *T.mu* group (Supplementary Fig. 6E–I). Overall, these results suggest that *T.mu* colonization can directly impact the *C. difficile*-host-community inter-related arginine/ornithine metabolism.

iNOS and ARG1 are used to define classically activated M1 (iNOS$^+$) and alternatively activated M2 (ARG1$^+$) macrophages. These two enzymes and their related metabolites are fundamentally involved in the intrinsic regulation of macrophage polarization and function[37]. We wondered whether *T.mu* colonization influences macrophage polarization after *C. difficile* infection. We, therefore, performed ARG1/[F4/80] and iNOS/[F4/80] co-immunostaining of colon tissues collected from *C. difficile*-infected WT mice. There was a reduction of ARG1$^+$F4/80$^+$ cells in the gut of *T.mu*-colonized mice post infection, while the iNOS$^+$F4/80$^+$ cells were similar between the *T.mu*-colonized and *T.mu*-uncolonized mice (Supplementary Fig. 7). Notably, iNOS expression was strongly induced at the intestinal villus tip epithelium of *T.mu*-colonized mice (Supplementary Fig. 7). These results suggest that *T.mu* colonization influence not only alternatively activated M2 macrophages but also other type of cells that are involved in the regulation of intestinal arginine/ornithine metabolism during *C. difficile* infection.

## Modulating arginine/ornithine metabolism is important for controlling *C. difficile* infection

To evaluate whether modulating arginine/ornithine metabolism can control CDI, we added extra arginine in the drinking water, or made an ornithine-free diet. When mice treated with 1% arginine in their drinking water or fed the ornithine-free diet, we found that, just like *T.mu* colonization did, arginine supplementation in water or ornithine depletion in diet increased arginine and decreased ornithine in the cecal contents, respectively, 48 h post *C. difficile* infection (Fig. 6A, B). Supplementation with arginine or ornithine-free diet significantly reduced *C. difficile*-induced disease severity (Fig. 6C–G). Furthermore, both arginine supplementation and ornithine-free diet decreased the recruitment of neutrophils and the expression of CXCL1and IL-1β in the intestine post CDI (Fig. 6H–K). Supplementation with arginine did not affect *C. difficile* colonization, but significantly reduced TcdB titers in the cecal contents (Fig. 6L–N). Notably, ornithine-free diet reduced both TcdB titers and *C. difficile* spore biomass (Fig. 6L–N). However, supplementation of 2% citrulline in the drinking water did not show improvement of either *C. difficile*-infection-induced symptoms or intestinal damage (Supplementary Fig. 8A–E). In addition, like *T.mu*-colonized mice, mice fed with ornithine-free diet had an increase of

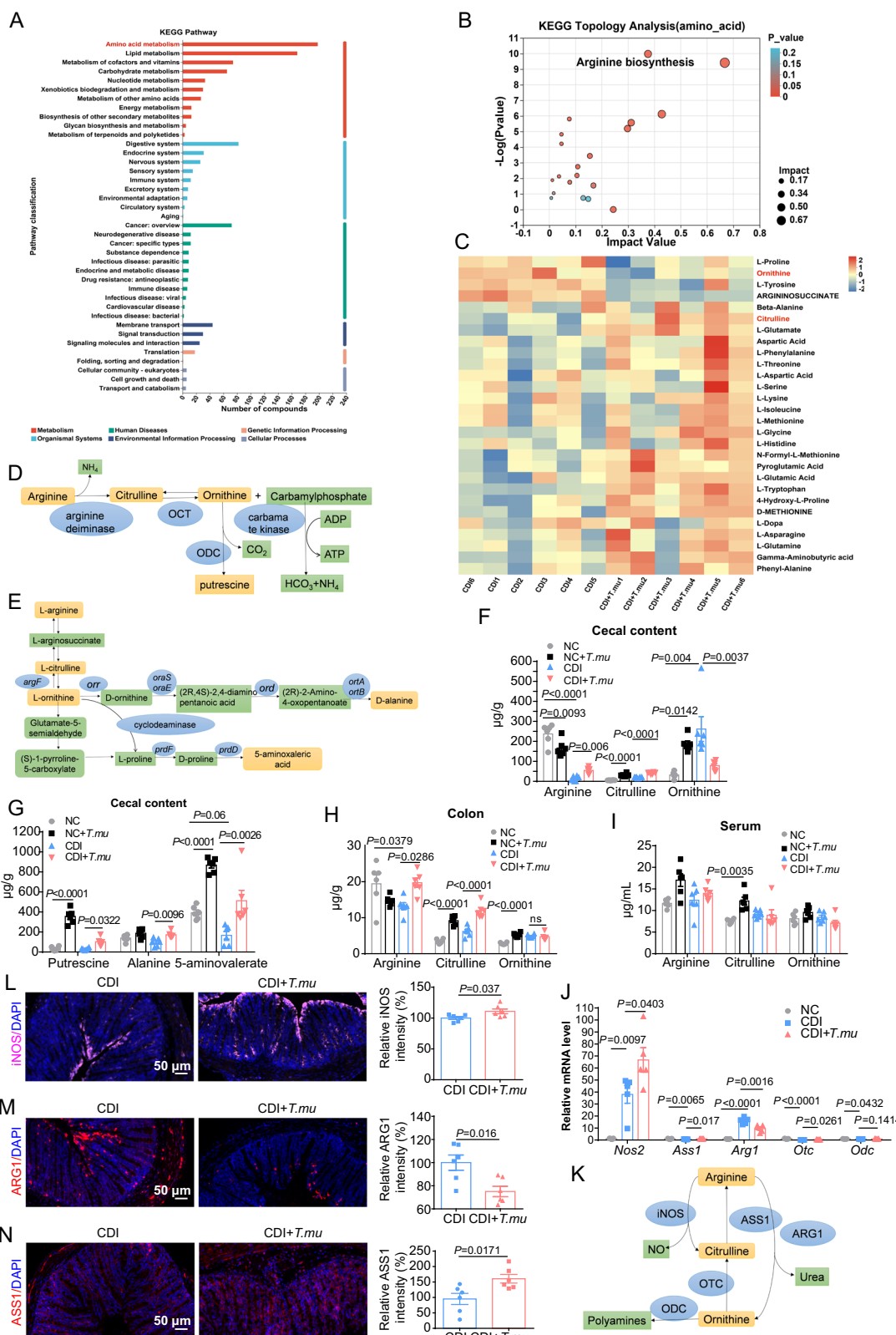

Th1 cells in the gut, and the IFN-γ concentration in the gut was also increased post CDI (Fig. 6O–Q). Together, these results suggest that modulating arginine/ornithine metabolism is important for controlling CDI.

Next, we wanted to explore the role of host arginine metabolism in the protective effect of *T.mu* in CDI. α-Methyl-DL-aspartic acid (α-MDLA) is an inhibitor of ASS1[38], a rate-limiting enzyme in the citrulline-arginine cycle. We administered α-MDLA to the *T.mu*-colonized CDI mice (labeled as "CT+α-MDLA" group) to block host arginine synthesis via citrulline (Fig. 7A). *T.mu*-colonized mice in the CDI + *T.mu* group had increased arginine in the cecal contents, compared to the CDI group. After the α-MDLA treatment, the *T.mu*-colonization-induced arginine increase was blocked (compare CDI + *T.mu* group with CT+α-MDLA group, Fig. 7B). In addition, the α-MDLA treatment also blocked

**Fig. 5 | *T.mu* colonization influence gut microbial community and host arginine/ornithine metabolism. A–H** The colon, cecum, cecal content, and peripheral blood samples from the indicated mice were collected at day 2 post *C. difficile* infection. The untargeted metabolomic profiling was conducted for the cecal content samples collected from the CDI and the CDI + *T.mu* groups. **A** Summary of the cecal content metabolomes enriched in various KEGG metabolic pathways. The ordinate is the secondary classification of KEGG metabolic pathway, and the abscissa is the number of metabolites annotated to this pathway (*n* = 6 per group). **B** KEGG pathway enrichment topology analysis with Relative-betweenness Centrality method. Each bubble represents a KEGG pathway, the horizontal axis represents the impact value of the relative importance of metabolites in the pathway and the vertical axis represents the enrichment significance (i.e., [−log₁₀(*P* value)]) of metabolite participation pathway (*n* = 6 per group). *P* values were adjusted by Benjamini and Hochberg method to control FDR. FDR-adjusted *P* < 0.05 was shown. **C** Heatmap of differentially enriched amino acids between the cecal content samples from CDI vs. CDI + *T.mu*. Each column in the figure represents a sample, and each row represents a metabolite. The color in the

figure represents the relative expression quantity of metabolites in this group of samples. For the specific change trend of expression quantity, please see the digital label under the color bar on the upper right (*n* = 6 per group). **D** Schematic of the arginine dihydrolase pathway in *Trichomonads*. **E** Annotated biosynthetic and degradative pathways of L-ornithine in *C. difficile*. **F** The levels of arginine, citrulline and ornithine in the cecal content of the indicated mice 2 days post *C. difficile* infection (*n* = 6 per group). **G** The levels of putrescine, alanine, and 5-aminovalerate in the cecal content of the indicated mice 2 days post CDI (*n* = 6 each group). **H, I** The levels of arginine, citrulline and ornithine in the **H** colon, and **I** serum of the indicated mice 2 days post CDI (*n* = 6 per group). **J** The relative expression levels of *Nos2*, *Ass1*, *Arg1*, *Otc*, and *Odc* in the colon were determined by qRT-PCR (*n* = 5 per group). **K** Diagram of the host's arginine metabolic pathways. **L–N** Colon tissues were collected at day 2 post infection, and stained with ARG1 (red), iNOS (purple), ASS1 (red) and DAPI (blue) (*n* = 6 per group). Experiments were repeated independently two times. Data are the mean ± SEM. Statistical significance was determined by Fishers' Exact test (**B**), one-way ANOVA (**F–I, J**) and two-sided Students' *t* test (**L–N**). ns no significance. Source data are provided as a Source Data file.

*T.mu*'s protective effect on body weight, disease index, colon pathology, and inflammatory cell infiltration post *C. difficile* infection (Fig. 7C–G). Furthermore, α-MDLA reversed the *T.mu*'s reducing effect on neutrophil recruitment and TcdB titers in the colon post infection (Fig. 7H–K). Together, these results suggest that *T.mu* colonization modulates host arginine metabolism to influence immune cells in the gut to temper susceptibility to *C. difficile* (Fig. 7L).

## Discussion

This study shows that commensal protozoan *T.mu* protects against CDI. On the one hand, *T.mu* colonization in the gut reduces the damage caused by CDI-induced excessive recruitment of neutrophils to the gut; on the other hand, it protects the intestine by regulating the production of IFN-γ by Th1 cells. This sheds light on the interaction between the commensal protozoan *T.mu*, the pathogen *C. difficile*, and the host immune system in the infection sites.

Amino acid metabolism plays an important role in protozoa, and the arginine dihydrolase pathway can provide energy for *Tritrichomonas spp.* growth by transforming arginine into ornithine and carbamylphosphate, which then converting to putrescine and CO₂ plus ammonia, respectively[29,39]. Our results showed that the *T.mu* cultured in a rich medium containing sugar (i.e., BHI broth) in vitro still consumed a significant amount of arginine and released citrulline and ornithine (Supplementary Fig. 5A–C), suggesting *T.mu* may be able to use arginine dihydrolase pathway to metabolize arginine. In contrast, *C. difficile* cultured in the same rich medium was less active in metabolizing arginine. Consistent with this, compared with normal WT control, *T.mu*-colonized normal B6 mice had reduced arginine level in the cecal contents, while the levels of citrulline, ornithine, and putrescine were increased (Fig. 5F, G). We speculate that the ability of *T.mu* to metabolize arginine and ornithine may be important for its protective role in CDI. The reason is related to the fact that ornithine is important nutrient utilized by *C. difficile* for its colonization. Ornithine oxidative metabolism in *C. difficile* supports its non-inflammatory asymptomatic colonization in mice[40]. In addition, the modulation of the availability of arginine and ornithine levels by *T.mu* are likely to be sensed by both the host and the gut microbial community, which can, in return, influence *C. difficile* functions. The cross-regulation between different microbial members is not rare to find. For examples, *Clostridium sardiniense* uses arginine dihydrolase pathway to produce ornithine, allowing it to be absorbed by *C. difficile* as a Stickland substrate, and therefore, promoting *C. difficile* growth and worsening CDI in the GF mice; while *Paraclostridium bifermentans* can also use arginine dihydrolase pathway to produce ornithine, but it can also consume ornithine via Stickland pathways for its own metabolism, depriving *C. difficile* of the important Stickland substrate and

therefore, protecting the host against lethal CDI[41]. *C. difficile*, like other cluster XI *Clostridia*, utilizes diverse carbon sources, including ornithine fermented via Stickland reactions, for its rapid growth[42]. *Enterococci* increase *C. difficile* fitness in the antibiotic-perturbed gut by providing fermentable amino acids, including leucine and ornithine, and promote *C. difficile* virulence by depleting intestinal luminal arginine[34]. How exactly *T.mu* colonization influences the host and the gut microbial community arginine/ornithine metabolism is a very complicated problem remained to be resolved.

We found that *T.mu* colonization reduced intestinal inflammatory responses and *C. difficile* toxicity in both the GF and the SPF mice while not affecting *C. difficile* colonization efficacy, resulting in a relatively low toxicity and low-inflammatory state. This is correlated with a relatively reduced ornithine level in gut lumen of *T.mu*-colonized mice after CDI (Fig. 5F and Supplementary Fig. 6A). In accordance with this, it has been reported that mice resistant to CDI and inflammation exhibit enhanced ornithine oxidative metabolism and have relatively lower ornithine level in the gut lumen[40]. Clearly, there is still much to be learned about the complex interplay between *T.mu* and *C. difficile*.

It should be mentioned that although our GF mice experiment strongly suggests a direct impact of *T.mu* on CDI, but since we have not been able established a method to grow *T.mu* axenically in vitro, there is still possibilities that the immunomodulatory phenotype observed in infected animals were mediated by *T.mu*-associated microbial symbiont, or from the composite of a protozoan–microbial endosymbiotic interaction. Regardless, we believe that the protist and any associated microbial symbiont should be treated as a single microbial functional group that possesses distinct immunomodulatory properties during CDI.

In our study, we focused more on the host-integrated response rather than just the crosstalk between the two microorganisms, *T.mu* and *C. difficile*. In addition to serving as a substrate for enzymes, arginine itself selectively regulates the expression of several enzymes in a concentration-dependent manner, thereby affecting its own metabolism[43]. Our results showed that *T.mu*-colonized mice upregulated intestinal *Nos2*, *Otc*, and *Ass1* expression, while downregulated *Arg1* expression post infection, compared to the *C. difficile*-infected *T.mu*-free control mice (Fig. 5J–N). These results overall suggest that *T.mu* colonization influences the host's intestinal arginine metabolism at least during CDI. To further demonstrate the role of host arginine metabolism in protecting against CDI by *T.mu*, we used a small molecular inhibitor α-MDLA to inhibit ASS1 bioactivity in *T.mu*-colonized mice, and the inhibitor blocked *T.mu*'s protective effect on CDI (Fig. 7). Therefore, the influence of *T.mu* on the host's metabolic activities plays an even more important role in determining the outcome of *C. difficile* infection.

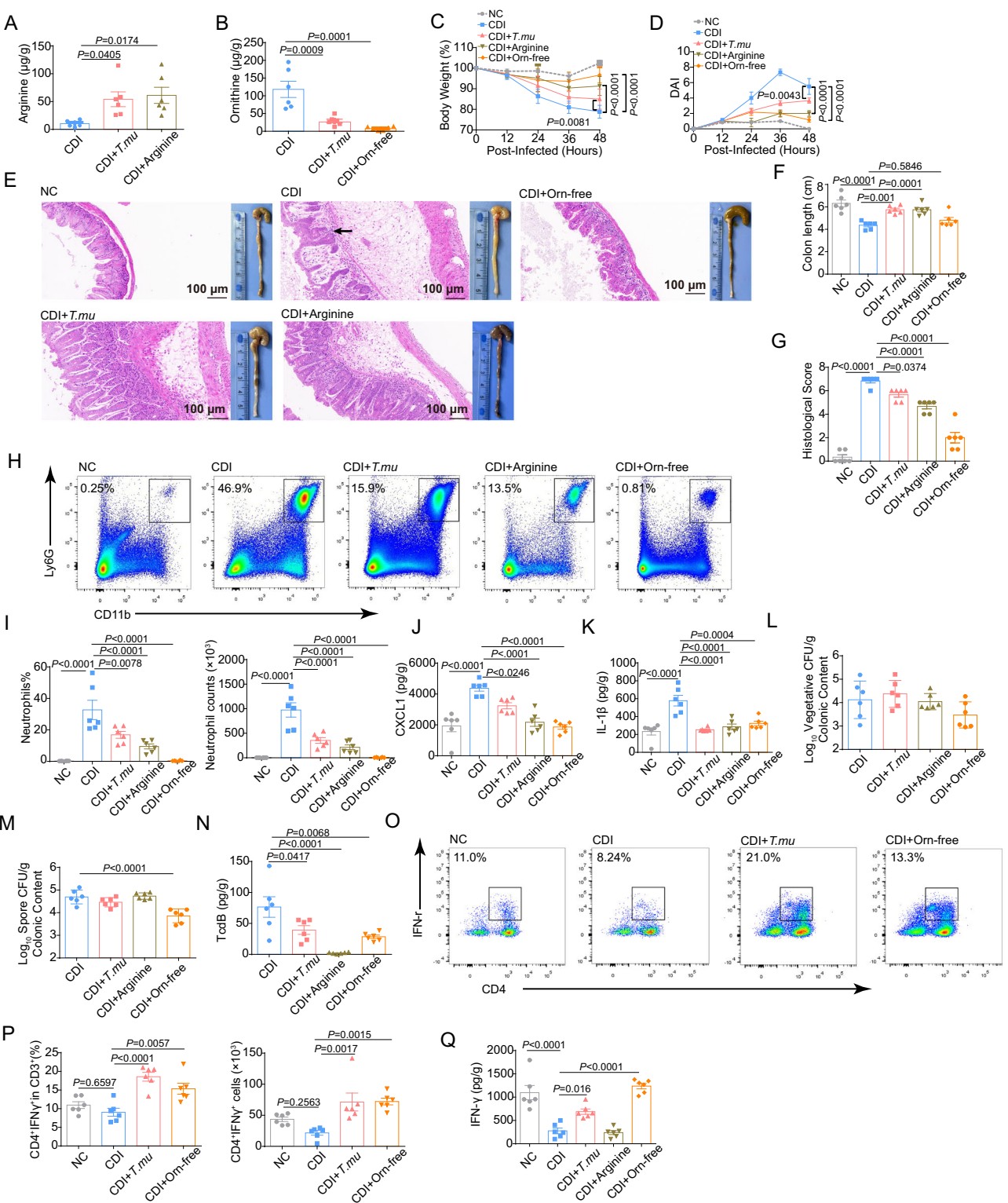

Our study also found that *T.mu* colonization alters the host intestinal immune environment, including inhibiting excessive infiltration of neutrophils (Fig. 3) and inducing intestinal Th1 cells to release IFN-γ (Fig. 4). This is correlated to *T.mu*'s inhibitory effect on CDI-induced expression of cytokines IL-1β, IL-6, and CXCL1, which are critical for the recruitment of neutrophils and tissue inflammation. Moreover, we found that the protective effect of *T.mu* in CDI disappeared in *Ifngr* knockout mice. *C. difficile* infection decreased the number of goblet cells and the production of MUC2 in the gut, *T.mu* colonization tempered this reduction in the WT mice but not in the

*Ifngr* KO mice. Our data showed a role of *T.mu* in the regulation of host immunity, and the host immunological tones plays a significant role in the progression and outcome of CDI[23,44].

For many years, dietary arginine supplementation has been used as a means to boost the immune system[32]. We speculated that the ability of *T.mu* to regulate host immunity might be related to arginine−ornithine metabolic pathway. Therefore, we supplemented WT B6 mice with arginine in drinking water or fed the mice with ornithine-free diet, and these treatments reduced intestinal neutrophil infiltration and IL-1β secretion post CDI. Furthermore, when the host's

**Fig. 6 | Modulating arginine/ornithine metabolism is important for controlling *C. difficile* infection. A–Q** From day −7 to day −1, mice in the CDI + *T.mu* group were administrated with *T.mu* every other day for a total of four times. Mice in the CDI +Orn-free group were given ornithine-free diet from day −7 to day 1. From day −3 to day 1, mice in the CDI +Arginine group were given 1% arginine in the drinking water. On day 0, each mouse in the CDI, CDI + *T.mu*, CDI +Arginine, and CDI +Orn-free groups was inoculated with *C. difficile* spores. The cecum and colon of the indicated mice were collected at day 2 post infection. **A, B** The levels of **A** arginine and **B** ornithine in the cecal contents (*n* = 6 per group). **C** Body weight changes post infection (*n* = 6 per group). **D** DAI (*n* = 6 per group). **E** Macroscopic photos of colon and representative HE-stained microscopic graph of cecal tissue sections. Scale bar: 100 μm. Arrow indicates infiltration of inflammatory cells. **F** Measurement of colon length (*n* = 6 per group). **G** Histological score for HE-stained cecal tissues (*n* = 6 per

group). **H, I** Representative flow cytometry graphs and the neutrophils percentages and numbers in the colonic lamina propria at 48 h post CDI (*n* = 6 per group). **J, K** The levels of **J** CXCL1 and **K** IL−1β in the cecum were determined by ELISA (*n* = 6 per group). **L−N** The numbers of *C. difficile* **L** vegetative cells, **M** spores, and the titers of (**N**) TcdB in cecal contents at 48 h post CDI (*n* = 6 per group). **O, P** Representative flow cytometry graphs and the percentages and numbers of Th1 cells in colonic lamina propria at 48 h post CDI (*n* = 6 per group). **Q** IFN-γ level in the cecum at 48 h post *C. difficile* infection was determined by ELISA (*n* = 6 per group). Experiments were repeated independently two times. Data are the mean ± SEM. Statistical significance was determined by two-way ANOVA (**C, D**) or one-way ANOVA (**A, B, F, G, I−N, P, Q**). Source data are provided as a Source Data file.

ASS1 activity was inhibited, the intestinal lumen arginine level was reduced, and *T.mu's* suppressive effect on CDI-induced neutrophil recruitment into the intestine was also diminished. In addition, when given ornithine-free diet to mice, Th1 cells in the gut were significantly expanded post *C. difficile* infection, promoting IFN-γ secretion and protecting the intestinal mucosa. These lines of evidence suggest that modulating intestinal arginine metabolism can influence the host immunity at least during CDI. In addition, previous reports indicate an effect of ornithine and arginine on *C. difficile* virulence[34,40,45,46]. Consistently, our data show that dietary manipulation of arginine/ornithine impacts *C. difficile* toxin levels. Thus, it is very likely that the interrelated arginine−ornithine metabolic axis regulates *C. difficile* infection outcome by coordinately influencing both the host immunity and the pathogen virulence.

In conclusion, our study focused on intestinal metabolism and immunity to explore the mechanism of commensal protozoan *T.mu* on intestinal inflammation induced by *C. difficile*. We propose a model of three-way interaction between a commensal protozoan, *C. difficile*, and the host immune system via arginine−ornithine metabolic axis to maintain intestinal homeostasis and relieve *C. difficile* infection. Understanding the molecular mechanisms by which a commensal eukaryotic member in tempering a pathogenic disease provides a better chance for understanding pathogenesis and developing refined therapeutic strategies.

## Methods
### Animals
All mouse studies were evaluated by the Laboratory Animal Ethics Committee of Xuzhou Medical University (IACUC number: 202202A278, Xuzhou, China). We made every effort to minimize animal suffering and to reduce the number of animals used. Wild-type male C57BL/6J mice, aged 4−6 weeks, were purchased from Xuzhou Medical University. The C57BL/6J *Ifngr*[−/−] male mice, aged 4−6 weeks, were provided by Professor Zhinan Yin of Jinan University (Guangzhou, China). The mice were bred and maintained under specific pathogen-free (SPF) condition. Male C57BL/6J germ-free mice, aged 4−6 weeks, were purchased from and housed under germ-free conditions in GemPharmatech Co., Ltd (Nanjing, China). All mice were housed on a 12-h alternating day and night cycle, and standard laboratory sterilized feed and water were freely available. The room temperature was 25 °C, and the humidity was 40−70%.

### Isolation and purification of *T.mu*
The cecal contents, collected from the *T.mu*-bearing mice, were suspended with sterile PBS, filtered several times through a 70-μm cell strainer (BS−70-CS, Biosharp Life Sciences, China), and washed three times with PBS. The pellet enriched with *T.mu* were resuspended in 40% percoll (17089109, cytiva, USA) to perform percoll gradient separation for obtaining *T.mu*. For in vivo administration, each mouse was inoculated with ~2 × 10⁶ *T.mu* via oral gavaged. For *T.mu* culture, the purified *T.mu* was then suspended with brain

heart infusion (BHI) broth (CM1135, OXOID, England) supplemented with a cocktail of broad-spectrum antibiotics, including 100 U/L penicillin (A6920, Solarbio, Beijing, China), 0.1 mg/mL streptomycin (A100382, Sangon Biotech, Shanghai, China), 50 μg/mL vancomycin (A600983, Sangon Biotech), 50 μg/ml ciprofloxacin (A600310, Sangon Biotech), 100 μg/ml gentamicin (A506614, Sangon Biotech), and 5 μg/mL amphotericin B (A610030, Sangon Biotech). After suspension, *T.mu* was then anaerobically incubated at 37 °C for 2 days and collected for gavage of germ-free mice.

### Scanning electron microscopy
According to a previous report[24], purified *T.mu* was suspended with fixative solution (2.5% glutaraldehyde + 4% paraformaldehyde in PBS) for 2 h and washed three times. Then samples were dehydrated using the following series ethanol-water mixtures: 25%, 50%, 70%, 80%, 90%, 100%, 100%. After treatment with 100% ethanol, the samples were incubated in isoamyl acetate twice. Then samples were dried in a critical-point dryer HCP-2 (Hitachi Ltd., Japan) and attached to the sample table, gold coating was performed in a high vacuum coating instrument (Leica EM ACE600, Germany). Finally, the samples were observed using a scanning electron microscope Teneo VS (FEI, USA).

### *C. difficile* spore preparation
*C. difficile* VPI 10463 (ATCC 43255) strain was presented by Xiaojun Song from Hangzhou Medical College, China. *C.difficile* spores were prepared as described previously[47]. *C. difficile* was grown at 37 °C on BHI agar plates supplemented with 0.05% L-cysteine (A600132, Sangon Biotech) in anaerobic conditions (AnaeroPack, MGC, Japan) for 2 days in jars. Then, *C. difficile* was cultured at 37 °C in BHI broth supplemented with 0.05% L-cysteine anaerobically for 5 days. Bacterial culture was harvested by centrifugation and washed with cold PBS at least three times. The pellets were suspended with PBS and heat treated for 10 min at 70 °C to remove vegetative cell debris and obtain *C. difficile* spores. The numbers of viable spores were plated on BHI agar plates supplemented with 0.1% L-cysteine, 0.5% yeast extract (A515245, Sangon Biotech), and 0.1% sodium taurocholate (A601143, Sangon Biotech) in anaerobic atmosphere in jars and recorded as CFU/mL.

### Mice model of *C. difficile* infection
First, the mice were pretreated with ABX, the antibiotic mixture of 0.4 mg/mL kanamycin (A506636, Sangon Biotech), 0.035 mg/mL gentamicin (A506614, Sangon Biotech), 0.035 mg/mL polymyxin E (A606495, Sangon Biotech), 0.215 mg/mL metronidazole (A600633, Sangon Biotech), and 0.045 mg/mL vancomycin (A600983, Sangon Biotech) added to drinking water. After 5 days, the mice were intraperitoneal injected with 10 mg/kg clindamycin (A600312, Sangon Biotech). A day later, mice were infected with 5 × 10⁶ CFU of *C. difficile* spores by oral gavage. Body weight and symptoms of disease (stool characteristics, weight loss, and decreased response to stimuli) were recorded every 12 h after *C. difficile* infection, and mortality was

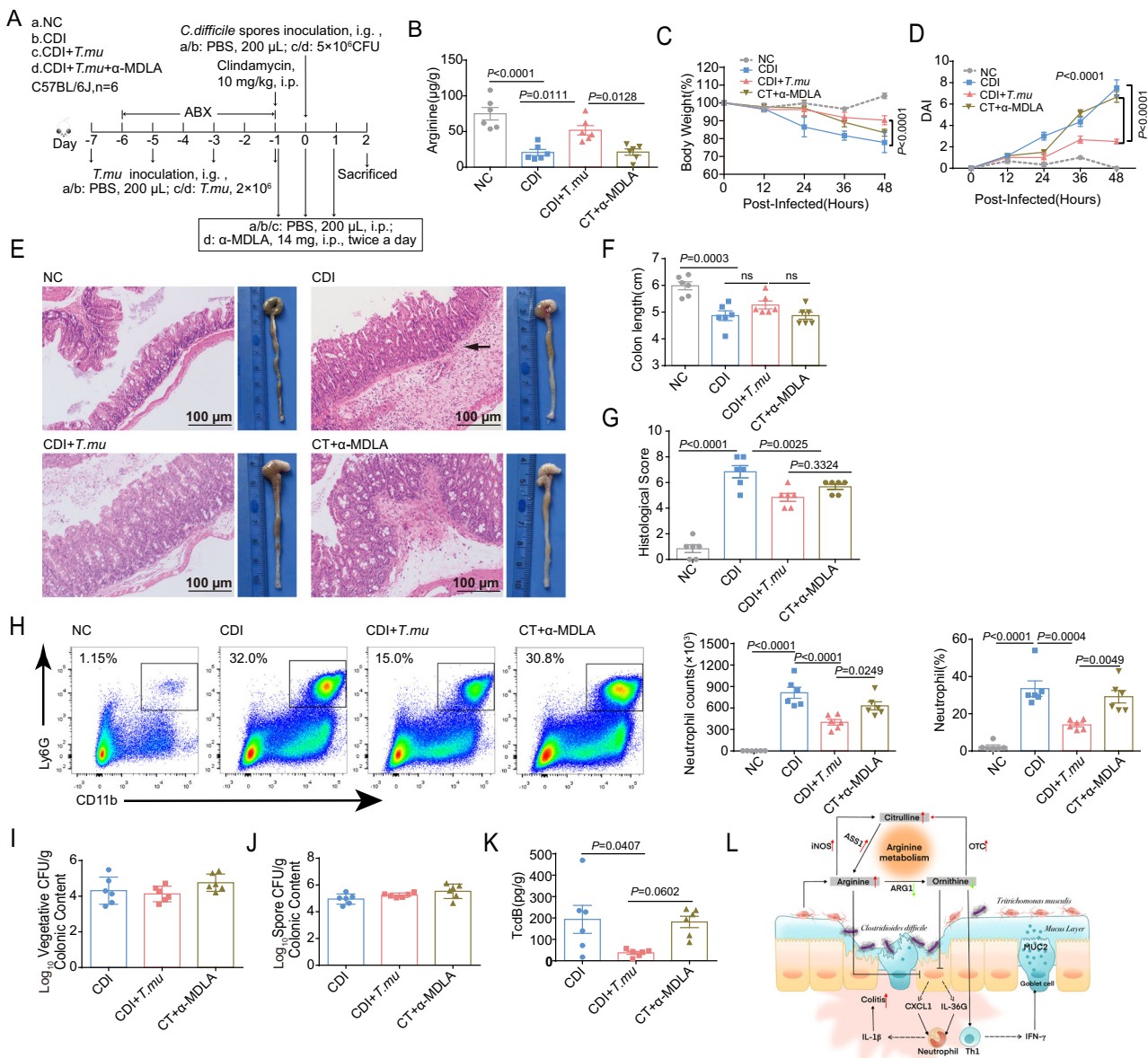

**Fig. 7 | α-MDLA prevented *T.mu* from relieving CDI. A–K** From days −7 to −1, mice in the CDI + *T.mu* and CT+α-MDLA groups were administrated with *T.mu* every other day for a total of four times. From day −1 to day 1, each mouse in the CT+α-MDLA group was intraperitoneally given 14 mg α-MDLA twice daily. On day 0, each mouse in the CDI, CDI + *T.mu*, and CT+α-MDLA groups was inoculated with *C. difficile* spores. The cecum and colon of the indicated mice were collected at day 2 post infection. **A** Schematic illustrating the experimental design. **B** The levels of arginine in the cecal contents (*n* = 6 per group). **C** Body weight changes post infection (*n* = 6 per group). **D** DAI (*n* = 6 per group). **E** Macroscopic photos of colon and representative HE-stained microscopic graph of cecal tissue sections. Scale bar: 100 μm. Arrow indicates infiltration of inflammatory cells. **F** Measurement of colon length (*n* = 6 per group). **G** Histological score for HE-stained cecal tissues (*n* = 6 per group). **H** Representative flow cytometry graphs and the neutrophils percentages and numbers in the colonic lamina propria (*n* = 6 per group). **I–K** The number of *C. difficile* **I** vegetative cells, **J** spores, and the titer of (**K**) TcdB in the cecal contents at 48 h post infection (*n* = 6 per group). **L** The hypothetical model of how *T.mu* influencing CDI. Experiments were repeated independently two times. Data are the mean ± SEM. Statistical significance was determined by two-way ANOVA (**C**, **D**), one-way ANOVA (**B**, **F–K**). ns no significance. Source data are provided as a Source Data file.

tracked. Disease activity index (DAI) was scored as described[48], varied from 0 (normal) to 12. Briefly, DAI is based on clinical symptoms of stool characteristics, behavioral change, and percent weight loss. Each category is scored from 0 to 4, and the individual values are added to provide an overall score.

### *T.mu* or drug treatment model

In the *T.mu-C. difficile* co-colonization model of SPF mice, mice were orally gavaged with $2 \times 10^6$ *T.mu* every other day in a week. For experiments, the mice were randomly divided into four groups: normal mice (NC group), normal mice treated with *T.mu* (NC + *T.mu* group), mice treated with PBS followed by *C. difficile* infection (CDI group), mice treated with *T.mu* followed by *C. difficile* infection (CDI + *T.mu* group).

Unlike SPF wild-type mice, co-colonized models of germ-free wild-type mice did not require antibiotic pretreatment. For experiments, the germ-free mice were divided into three groups: normal mice (NC group), mice treated with PBS followed by *C. difficile* infection (CDI group), mice treated with *T.mu* followed by *C. difficile* infection (CDI + *T.mu* group). First, CDI + *T.mu* group were inoculated with $2 \times 10^6$ *T.mu* for once, meanwhile CDI group were treated with equal PBS. 7 days later, both CDI group and CDI + *T.mu* group were orally gavaged with $5 \times 10^6$ CFU *C. difficile* spore. After a 36-h observation period, the feces, cecum, cecal contents, colon and colonic contents of

the mice were collected for further analysis. In addition, five mice in both CDI group and CDI + *T.mu* group were kept for long-term survival observation.

In the *T.mu* elimination experiment, the wild-type C57BL/6 J mice were divided into four groups: normal mice (NC group), mice treated with PBS followed by *C. difficile* infection (CDI group), mice treated with *T.mu* followed by *C. difficile* infection (CDI + *T.mu* group), and the elimination of *T.mu* colonization followed by *C. difficile* infection (CDI + *T.mu* + MNZ group). First, CDI + *T.mu* group and CDI + *T.mu* + MNZ group received an administration of $2 \times 10^6$ *T.mu*, meanwhile, CDI group were treated with equal PBS. A week later, CDI + *T.mu* + MNZ group were treated with 3 mg/mL metronidazole (MNZ) for 7 days, while 9 days later, CDI group and CDI + *T.mu* group were treated with ABX as described above for 5 days. Then mice were intraperitoneal injected with 10 mg/kg clindamycin. A day later, mice were infected with $5 \times 10^6$ CFU of *C. difficile* spores by oral gavage.

To investigate the role of IFN-γ in our model, *Ifngr*$^{-/-}$ mice were introduced into this experiment, mice were divided into four groups: normal mice (NC group), mice treated with PBS followed by *C. difficile* infection (CDI group), mice treated with *T.mu* followed by *C. difficile* infection (CDI + *T.mu* group), *Ifngr*$^{-/-}$ mice treated with *T.mu* followed by *C. difficile* infection (CT+Ifngr$^{-/-}$ group). First, mice were orally gavaged with $2 \times 10^6$ *T.mu* every other day in a week. Then mice were intraperitoneal injected with 10 mg/kg clindamycin followed by *C. difficile* infection. Besides, we also performed the IFN-γ recombinant protein (rIFN-γ) intervention experiment, each mouse in the rIFN-γ intervention group (CDI+rIFN-γ group) was intraperitoneally injected with 10 μg rIFN-γ recombinant protein (50709-MNAH, SinoBiological, China) at 2 h and 24 h after *C. difficile* infection.

To study the role of several amino acids in our model, mice were divided into five groups: normal mice (NC group), mice treated with PBS followed by *C. difficile* infection (CDI group), mice treated with *T.mu* followed by *C. difficile* infection (CDI + *T.mu* group), mice treated with 1% arginine (Sangon Biotech) 3 days before *C. difficile* infection (CDI+Argine group), mice treated with 2% citrulline (Sangon Biotech) 7 days before *C. difficile* infection (CDI+Cit group), and mice treated with ornithine-free diet (Jiangsu Xietong Pharmaceutical Bio-engineering Co., Ltd., Jiangsu, China) for 7 days before *C. difficile* infection (CDI+Orn-free group). This diet formula refers to the published literature[40].

To study the role of argininosuccinate synthase 1 (ASS1) in our model, the ASS1-specific inhibitor, α-Methyl-DL-aspartic acid (α-MDLA, HY-W142119, MCE, USA) intervention experiment was performed. The mice were randomly divided into four groups: normal mice (NC group), mice treated with PBS followed by *C. difficile* infection (CDI group), mice treated with *T.mu* followed by *C. difficile* infection (CDI + *T.mu* group), mice treated with *T.mu* and α-MDLA followed by *C. difficile* infection (CT+α-MDLA group). First, CDI + *T.mu* group and CT+α-MDLA group were orally gavaged with $2 \times 10^6$ *T.mu* every other day in a week. Then mice were intraperitoneal injected with 10 mg/kg clindamycin followed by *C. difficile* infection. Meanwhile, each mouse in the CT+α-MDLA group was intraperitoneally inoculated with 14 mg α-MDLA twice daily for 3 days.

### Quantification of *T.mu* in the feces or cecal content
The cecum of mice was cut longitudinally and weighed. The fresh feces or harvested cecal contents were suspended in PBS (10 μL/mg). The protist was counted using a hemocytometer as previously reported, total number and concentration of *T.mu* were calculated accordingly.

### Quantification of *C. difficile* and toxins
After 48 h post *C. difficile* infection, mice were sacrificed, and the cecum and colon, as well as the feces, cecal contents, and colonic contents, were removed aseptically, weighed, and homogenized in

1 mL of PBS containing 0.3% Triton X−100 (A600198, Sangon Biotech), serially diluted and plated on CCFA agar (HB8808, Hopebio, Shandong, China) plates supplemented with 50% egg yolk emulsion (HB8295, Hopebio), 0.5 mg/mL D-cycloserine (HB0254, Hopebio) and 16 μg/mL cefoxitin (C859229, Macklin, Shanghai, China) for enumeration of the vegetative cells. To quantify *C. difficile* spores, samples were further heated at 70 °C for 10 min, and coated on the CCFA agar plates supplemented with 50% egg yolk emulsion, 0.5 mg/mL D-cycloserine, 16 μg/mL cefoxitin, 0.5% yeast extract and 0.1% sodium taurocholate. Level of *C. difficile* toxins B in cecal contents and feces were measured by ELISA using the Mouse CDT-B ELISA KIT (ml058147, mlbio, Shanghai, China).

### Gross and histological assessment
The whole colon from the end of the cecum to the anus was photographed and the length of the colon was measured. For histological analysis, the cecum was fixed in 4% paraformaldehyde (A500684, Sangon Biotech) for 48 h and embedded into paraffin. Tissue sections (4 μm) were stained with hematoxylin and eosin. Images were captured with a microscope (DP74, Olympus, Japan). The histological score was evaluated according to the criteria described previously[48]. Briefly, histologic injury was graded based on epithelial tissue damage, amount of edema, and neutrophil infiltration. Each category was scored from 0 to 3 with the individual values added for an overall score.

### Goblet cell staining
Paraffin-embedded tissue sections (4 μm) were dewaxed with xylene (10023418, SINOPHARM, Shanghai, China) and 100%/100%/70% ethanol (100092683, SINOPHARM, Shanghai, China), respectively. After dewaxing, sections were stained with PAS dye solution B (G1049, Servicebio, Hubei, China) for 15 min, then PAS dye solution A for 30 min in the dark and PAS dye solution C for 30 s. Finally, the slides were soaked in anhydrous ethanol for three times to dehydrate, soaked in xylene for two times to transparent. Images were captured using an Olympus DP74 microscope.

### Immunohistochemistry
For immunohistochemical analysis, paraffin-embedded tissue sections (4μm) were deparaffinized, dehydrated, and subjected to antigen retrieval in citrate buffer (pH 6.0). The sections were then exposed to 3% hydrogen peroxide to block endogenous peroxidase activity, blocked with 5% bovine serum albumin (BSA, A8010, Solarbio, Beijing, China) for 30 min, and stained with anti-MUC2 antibody (27675-1-AP, Proteintech, USA, 1:1000 dilution) overnight at 4 °C. The sections were washed at least three times with PBS and stained with horseradish peroxidase (HRP)-labeled secondary antibody (GB23303, Servicebio, 1:200 dilution) for 1 h and finally stained with hematoxylin, dehydrated, and mounted. Images were captured using an Olympus DP74 microscope. Staining intensity was analyzed using the ImageJ software.

### Immunofluorescence
Paraffin-embedded tissue sections (4 μm) were soaked in xylene for three times, then rehydration through an ethanol to water gradient at 100%/100%/95%/ 90%/80%/70%. Then, sections were placed in Citrate Antigen Retrieval Solution (0.1 M, pH 6.0) and heated for 5 min. After cooling to room temperature (RT), sections were blocked with 5% BSA for 1 h, then incubated with the anti-arginase antibody (ab233548, EPR22033-369, abcam, England, 1:1000 dilution) and anti-ASS1 antibody (ab170952, EPR12398, abcam, 1:1000 dilution) at 4 °C overnight, or incubated with the anti-iNOS antibody (ab283655, RM1017, abcam, 1:200 dilution) and anti-F4/80 antibody (ab300421, EPR26545-166, abcam, 1:10,000 dilution) at 37 °C for 1 h. After that, sections were washed five times and then incubated with the appropriate secondary antibody (PV-6001, ZSGB-BIO, Beijing, China) for 20 min at RT. Finally,

the tissue sections were counterstained with DAPI (C1005, Beyotime) for nuclei visualization. Images were acquired using a scanister (Tissue Gnostics, Austria). To allow comparison between groups, fluorescence intensity was measured using ImageJ software.

## Enzyme-linked immunosorbent assay (ELISA)

The cecum of mice was removed and homogenized in RIPA lysis buffer (KGB5203, KeyGEN Bio TECH, Jiangsu, China) containing protease inhibitor cocktail (CW2200, Cwbio, China). Homogenates were incubated on ice for 30 min, then centrifuged at 10000×$g$ for 10 min, supernatants were collected and used for measurement of IL-1β, IL-17A, IL-6, IL-13, IFN-γ, IL-4, IL-36G and CXCL1. IL-1β, IL-13, IL-4, IL-17A, and IFN-γ were measured by Mouse Uncoated ELISA Kit (Invitrogen, USA) according to the manufacturer's instructions. IL-36G was quantified using Mouse ELISA kits (DL-IL1F9-Mu, DLdevelop, Jiangsu, China), and CXCL1 was quantified using Mouse ELISA kits (EM0003, Fine test, Hubei, China) according to the manufacturer's instructions.

## RNA extraction and qRT-PCR

RNA was extracted from colonic tissues using the Total RNA Extraction Kit (R1200, Solarbio, Beijing, China) in accordance with the manufacturer's protocols. RNA was converted to cDNA using the PrimerScript RT Reagent Kit (RR037A, Takara, Japan). After reverse transcription, qRT-PCR was performed using SYBR Green qPCR Master mixes (b21203, Bimake, USA) on the 7900 Fast Real-Time PCR system (Roche, Switzerland). The PCR program was as follows: 95 °C for 10 min, followed by 40 cycles of 95 °C for 10 s, 60 °C for 30 s, and 72 °C for 32 s. $2^{-\Delta\Delta CT}$ method was used to calculate the relative gene expression. β-actin was used as the endogenous control, and transcription levels of indicated genes were normalized to it. The primers used are listed in Supplementary Data 1.

## Lymphocytes preparation

To isolate lymphocytes from the colonic lamina propria, the colons were opened longitudinally, washed with PBS to remove luminal feces, and cut into 1-cm pieces, followed by shaking in cold PBS and then incubating with cold PBS containing 10 mM EDTA at 200 rpm and 37 °C for 30 min to remove the epithelial cells. Next, the lamina propria tissues were sliced with pieces and digested with the RPMI 1640 medium containing 5% fetal bovine serum (FBS, HY-T1000, ExCell Bio, Uruguay), 100 U/L penicillin, 0.1 mg/mL streptomycin, 1 mg/mL collagenase (11088866001, Sigma-Aldrich, USA), 1 mg/mL hyaluronic acid (935166, Sigma-Aldrich, USA), and 1 μg/mL DNase I (D806930, MACKLIN) at 100 rpm and 37 °C for 1 h. After incubation, the digested solution was filtered through a 70-μm cell strainer to obtain single-cell suspensions and resuspended in 40% percoll, followed by centrifugation at 670 × $g$ for 30 min at 4 °C to perform percoll gradient separation for obtaining lamina propria lymphocytes. The cell pellets were washed with cold PBS and resuspended in PBS containing 2% FBS. Spleens were directly ground into single-cell suspensions and disposed with erythrocyte lysis buffer to obtain splenic lymphocytes. Bone marrows were obtained from long bones of sacrificed mice.

## Flow cytometry

For neutrophils and macrophages analysis, single-cell suspensions were stained for 30 min at 4 °C in the dark with the following antibodies: anti-CD45 conjugated to PE-Cyanine7 (147704, I3/2.3, Biolegend, USA, 1:200 dilution), anti-CD11b conjugated to AlexPacific Blue (101224, M1/70, Biolegend, 1:200 dilution), anti-Ly6G conjugated to PE (127607, 1A8, Biolegend, 1:200), and anti-F4/80 conjugated to FITC (123108, BM8, Biolegend, 1:200 dilution).

For analysis of Th and Tc cells, 2 × 10⁶ cells were stimulated with cell activation cocktail (1:500 dilution, 423303, Biolegend) for 5 h at 37 °C in 5% CO₂. After incubation for 5 h, cells were washed with PBS and stained with Zombie NIR Fixable Viability Kit (423105, Biolegend,

1:200 dilution) for 10 min, then stained with anti-CD45 conjugated to PE-Cyanine7 (147704, I3/2.3, Biolegend, 1:100 dilution), anti-CD3 conjugated to PerCP/Cyanine5.5 (100328, 145-2C11, Biolegend, 1:100 dilution), anti-CD4 conjugated to FITC (100510, RM4-5, Biolegend, 1:100 dilution), anti-CD8 conjugated to APC (100711, 53-6.7, Biolegend, 1:100 dilution) for 40 min at 4 °C in the dark. Then cells were fixed with cell fixation buffer (420801, Biolegend) for 30 min at 4 °C, permeabilized with intracellular staining perm wash buffer (421002, Biolegend) and stained with anti-IFN-γ conjugated to PE (505808, XMG1.2, Biolegend, 1:100 dilution), anti-IL-17A conjugated to Brilliant Violet 421 (506925, TC11-18H10.1, Biolegend, 1:100 dilution).

For analysis of innate lymphoid cells (ILCs), 2 × 10⁶ cells were stained with anti-lineage (CD45R, CD11c, Gr-1, TCR β chain, TCR γ/δ, Fc εR1α, CD4, F4/80) conjugated to FITC (103205, 117305, 108405, 109205, 118105, 134305, 100510, 123108, Biolegend, 1:100 dilution), anti-CD45 conjugated to APC/Fire 750 (103153, 30-F11, Biolegend, 1:100 dilution), anti-CD335 (NKp46) conjugated to Brilliant Violet 711 (137621, 29A1.4, Biolegend, 1:100 dilution), anti-CD127 (IL-7Rα) conjugated to PE/Cyanine7 (135013, A7R34, Biolegend, 1:100 dilution) for 50 min at 4 °C. After staining with surface markers, cells were fixed for 1 h by True Nuclear Fix solution (424401, Biolegend), then stained with anti-T-bet conjugated to Brilliant Violet 605 (644817, 4B10, Biolegend, 1:20 dilution), anti-GATA3 conjugated to PE (653803, 16E10A23, Biolegend, 1:20 dilution), anti-EOMES conjugated to Alexa Fluor 647 (157703, W17001A, Biolegend, 1:200 dilution), anti-ROR gamma (t) conjugated to PerCP-eFluor 710 (46-6981-80, B2D, eBioscience, 1:100 dilution) in the True Nuclear Perm buffer (424401, Biolegend) for 1 h. Flow cytometry was performed on the FACS Canto II Flow Cytometer (BD Bioscience, USA), and analyzed by the FlowJo software.

## In vitro co-culture

To explore the effect of *T.mu* on neutrophil stimulated by *C. difficile*, neutrophils were purified from bone marrow-derived single-cell suspensions following double-gradient centrifugation using Histopaque −1119 and Histopaque-1077 (50/50 v/v; Sigma, USA). In total, 1 × 10⁵ cells were cultured at 37 °C in DMEM medium (VCM12008, VICMED, Jiangsu, China) containing 10% FBS. Initially, cells were incubated with 2 × 10⁶, 1 × 10⁷ or 5 × 10⁷ purified *T.mu* for 2 h. Then 1 × 10⁷ CFU/mL *C. difficile* were added to the co-culture for next 4 h. Subsequently, the supernatant was collected for measurement of IL-1β released in the culture.

To explore the effects of *T.mu* on the number of *C. difficile* and amino acids in vitro, 2 × 10⁵ CFU *C. difficile* were cultured with or without 2 × 10⁵ or 2 × 10⁶ *T.mu* in BHI broth. Cultures were anaerobically incubated for 24 h at 37 °C, then plated on BHI agar plates for *C. difficile* count and detected by mass spectrometry for amino acids quantification.

## Quantification of amino acids

Cultures, serum, cecal contents and colon tissues were suspended in 1 mL of extraction solvent (acetonitrile:methanol:water = 2:1:1) containing 0.1% formic acid followed by shaking, homogenizing, and sonicating, then the suspensions were incubated at −20 °C. After 1 h, the suspensions were centrifuged at 12,000×$g$ for 15 min, and supernatants were collected for quantification of arginine, citrulline, ornithine, putrescine, alanine and 5-aminovalerate by Liquid chromatography–mass spectrometry (LC–MS).

## Untargeted metabolomics analyses

LC/MS-based metabolomics were performed by Majorbio Biotech (Shanghai, China). CDI group ($n = 6$), CDI + *T.mu* group ($n = 6$), QC group (by mixing equal volumes of all samples from CDI group and CDI + *T.mu* group ($n = 3$)) was analyzed. Briefly, a 50 mg cecal contents sample was accurately weighed, and the metabolites were extracted using 400 μL of methanol: water (4:1, v/v) solution. The mixture was

allowed to settle at −10 °C and treated with a high-throughput tissue crusher Wonbio-96c (Shanghai Wanbo Biotechnology Co., Ltd, Shanghai, China) at 50 Hz for 6 min, followed by vortex for 30 s and ultrasound at 40 kHz for 30 min at 5 °C. The samples were placed at −20 °C for 30 min for precipitating the proteins. After centrifugation at 13,000 × $g$ and 4 °C for 15 min, the supernatant was carefully transferred to sample vials for LC−MS analysis.

Chromatographic separation of the metabolites was performed using a Thermo UHPLC system equipped with an ACQUITY UPLC HSS T3 column (100 mm × 2.1 mm i.d., 1.8 μm; Waters, Milford, USA). Mass spectrometric data were collected using a Thermo UHPLC-Q Exactive HF-X Mass Spectrometer equipped with an electrospray ionization source operating in either a positive or negative ion mode. The optimal conditions were set as follows: source temperature at 425 °C; Sheath gas flow rate at 50 arb; Aux gas flow rate at 13 arb; ion-spray voltage floating (ISVF) at −3500 V in negative mode and 3500 V in positive mode, respectively; Normalized collision energy, 20−40−60 V rolling for MS/MS. Full MS resolution was 60,000, and MS/MS resolution was 7500. Data acquisition was performed with the Data-Dependent Acquisition (DDA) mode. The detection was carried out over a mass range of 70−1050 $m/z$.

The pretreatment of LC/MS raw data were performed by Progenesis QI (Waters Corporation, Milford, USA) software, and a three-dimensional data matrix in CSV format was exported. The information in this three-dimensional matrix included: sample information, metabolite name, and mass spectral response intensity. Internal standard peaks, as well as any known false positive peaks (including noise, column bleed, and derivatized reagent peaks), were removed from the data matrix, deredundant and peak pooled. At the same time, the metabolites were identified by searching database, and the main databases were the HMDB, Metlin (https://metlin.scripps.edu/) and Majorbio Database.

The data were analyzed through the free online platform of Majorbio cloud platform (www.cloud.majorbio.com). Metabolic features detected at least 80% in any set of samples were retained. After filtering, minimum metabolite values were imputed for specific samples in which the metabolite levels fell below the lower limit of quantitation and each metabolic features were normalized by sum. To reduce the errors caused by sample preparation and instrument instability, the response intensity of the sample mass spectrum peaks was normalized by the sum normalization method, and the normalized data matrix was obtained. At the same time, variables with relative standard deviation (RSD) >30% of QC samples were removed, and log10 logarithmization was performed to obtain the final data matrix for subsequent analysis.

The KEGG PATHWAY database (Kyoto Encyclopedia of Genes and Genomes, http://www.genome.jp/kegg/)[49] is a collection of manually mapped metabolic pathways that describe molecular interactions, physiological and biochemical reactions, and relationships among gene products. According to the information of metabolite alignment to KEGG compound ID, the metabolic pathway information involved in metabolites can be obtained, so as to evaluate their impact on the biological metabolic process. The enrichment analysis of KEGG pathway refers to the enrichment analysis of the selected metabolic set, and the hypergeometric distribution algorithm is used to obtain the pathway of significant enrichment of metabolites in the metabolic set. In general, BH method is used by default to correct the $P$ value, and when the corrected $P$ value is <0.05, it is considered that there is significant enrichment in this pathway. KEGG topological analysis using Relative-betweeness centrality.

Metabolites with similar expression patterns are usually functionally related. The metabolites in the selected metabolic concentration were analyzed by cluster analysis, including cluster heatmap and subcluster trend chart. Hierarchical clustering was used for sample and amino acid clustering analysis. The hierarchical clustering method was Complete, and the distance algorithm was Euclidean.

## Microbiome analysis

Total DNA was extracted from cecal contents samples by using the DNeasy PowerSoil Pro Kit (QIAGEN, USA) according to the manufacturer's protocol. The concentration and purity were measured using a NanoDrop2000 (Thermo Fisher Scientific, USA). The 16 S rRNA genes of distinct regions V3−V4 were amplified using the primers 338 F (5′-ACTCCTACGGGAGGCAGCAG-3′) and 806 R (5′-GGAC-TACHVGGGTWTCTAAT-3′). The polymerase chain reaction (PCR) cycling conditions were as follows: 3 min at 95 °C for initialization, 30 cycles of 30 s denaturation at 95 °C, 30 s annealing at 52 °C, and 42 s extension at 72 °C, followed by a final elongation at 72 °C for 10 min. The length and concentration of the PCR products were determined by agarose gel electrophoresis. PCR products were mixed in equidensity ratios using the GeneTools Analysis Software (Version 4.03.05.0, SynGene). The PCR products were purified using the AxyPrep DNA Gel Extraction Kit (Axygen, USA). Sequencing libraries were generated using the NEXTFLEX Rapid DNA-Seq Kit for Illumina (Bioo Scientific, USA) as per the manufacturer's protocol. The library was sequenced on an Miseq PE300/NovaSeq PE250 platform (Shanghai Majorbio Technology Co., Ltd. Shanghai, China). Using fastp software (https://github.com/OpenGene/fastp, version 0.20.0) to quality control of original sequencing sequence, use FLASH software (http://www.cbcb.umd.edu/software/flash, version 1.2.7) splicing. Based on the default parameters, DADA2 plug-in in Qiime2 flow (or Deblur plug-in) is used to reduce the noise of the optimized sequence after quality control concatenation. Sequences after DADA2 denoising are often referred to as ASVs (amplicon sequence variants). Remove all samples annotated to chloroplast and mitochondrial sequences. To minimize the impact of sequencing depth on the subsequent analysis of Alpha diversity and Beta diversity data, the sequence number of all samples was scaled to 20,000. After the scaling, the average sequence coverage of each sample could still reach 99.09%. A species taxonomic analysis of ASVs was performed using Naive Bayes classifiers from Qiime2 based on the Sliva 16 S rRNA gene database (v 138). Bioinformatic analysis of the gut microbiota was carried out using the Majorbio Cloud platform (https://cloud.majorbio.com).

The principal coordinate analysis (PCoA) was conducted using ASV and samples to identify potential principal components that affect the differences in sample community composition by dimensionality reduction. Qiime (http://qiime.org/install/index.html) was used to calculate the distance of beta diversity based on Bray-Curtis distance metrics. Hierarchical clustering was performed according to the beta diversity distance matrix. The PCoA (Principal Co-ordinates Analysis) by R-3.3.1 (vegan). The ANOSIM (Analysis of similarities) was performed by R for PCoA of all groups.

The Heatmap used color gradients to represent the size of data in a two-dimensional matrix or table and presents information about community species composition. Hierarchical clustering was usually carried out according to the similarity of abundance among species or samples, and the results are presented on the community heatmap.

The composition proportion of dominant species in each group and distribution proportion of the dominant family in different groups were visualized with python-2.7 vegan package and Circos tool.

The linear discriminant analysis (LDA) effect size (LEfSe) (http://huttenhower.sph.harvard.edu/LEfSe) was performed to identify the significantly abundant taxa (family to family) of bacteria among the different groups (LDA score >2, $P$ < 0.05).

## Statistical analysis

Data are presented as the mean ± SEM. The statistical significance of the differences between groups was calculated by the nonparametric

Mann–Whitney test, the Kruskal Wallis test or two-way ANOVA analysis of variance using the SPSS 20 software (IBM Corp., Aemonk, NY, USA).

## Reporting summary

Further information on research design is available in the Nature Portfolio Reporting Summary linked to this article.

## Data availability

The Mass spec-based metabolomics data generated in this study have been deposited in the MetaboLights database (MTBLS9719, https://www.ebi.ac.uk/metabolights/MTBLS9719). The 16 S rRNA-seq data have been deposited in the NCBI Sequence Read Archive database SRA, PRJNA1077800. All remaining data generated or analyzed in this study are provided in the Source Data file. Source data are provided with this paper.

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

## Acknowledgements
The authors thank Xiaojun Song for sharing *C. difficile* VPI 10463 (ATCC 43255) strain. Funding was provided by the National Natural Science Foundation of China Grant 82102408 (H.Y.), 81871734 (B.G.), 82072380 (B.G.), China Postdoctoral Science Foundation Grant 2022M712681 (H.Y.), Jiangsu Provincial Natural Science Foundation Grant BK20231170 (H.Y.), advanced talents of Guangdong Provincial People's Hospital Research Foundation Grant KJ012021097 (B.G.) and Xuzhou Medical University Excellent Talent Introduction Project Grant D2019030 (H.Y.), D2020060 (X.L.).

## Author contributions
H.Y., X.W., Y.K. and Y.W. designed the study and prepared the manuscript. H.Y., W.Z., X.W., X.L., W.Z., Z.Z., L.W., Z.H., Q.C. and A.H. performed and analyzed experiments. X.W., H.Y., X.L., L.W., Y.Z., Y.C., Z.Y. and Y.K. contributed intellectually to the analysis and interpretation of the data. X.W., W.C. and H.Y. wrote the manuscript. H.Y., B.G. and X.L. provided funding for the research.

## Competing interests
The authors declare no competing interests.
