## [Peer Review File · Nature Communications]

A commensal protozoan attenuates *Clostridioides difficile* pathogenesis in mice through regulating the arginine-ornithine metabolic axis and the host intestinal immune responseREVIEWER COMMENTS

Reviewer #1 (Remarks to the Author):

In this study, the authors examine how the protozoan parasite Tmu affected disease progression following infection with Cdiff. They show that Tmu itself develops a symbiotic relationship in mice and doesn't affect the intestinal barrier or cause pathology. When mice are pre-colonised with Tmu before Cdiff infection, the authors find that Tmu in the gut alleviates overall disease scores without affecting Cdiff burden. They correlate this with reduced PMN recruitment, a boost in IFNG production and reduced arginase expression, which they theorise explains some of the metabolite changes they observe in the intestines.

This is an interesting paper that raises multiple questions. It is, however, largely descriptive in its findings, and there are many places where it isn't clear how phenotypes are linked or connected. It also isn't clear how Tmu is mediating any of the observed changes – there is no analysis on microbiome or experiments to address indirect effects. My specific comments and suggestions below.

1. My main comment for the authors is to address how Tmu is affecting these changes they observe. For example, what is the phenotype of germ-free mice when colonised with Tmu before Cdiff infection? Is it the same/similar effect? What happened to microbiome composition in Tmu colonised mice? Did it change – is that relevant? How do they know the metabolite changes they see were not caused by significant population changes occurring elsewhere? This would help us understand whether the effects of Tmu are direct on the host or indirectly mediated.

2. What were the numbers of neutrophils in the lamina propria? Authors show frequency (which is striking), but that could reflect significant expansions of other cell types in the Tmu mice, without affecting neutrophil number. A more comprehensive analysis of the cellular composition of Tmu mice is needed.

3. Related to above – the change in neutrophils in the gut is really striking. Was there any related changes to neutrophil production and circulation in the bone marrow? How are

these effects on PMNs related to the arginine metabolism – which phenotype comes first?

4. What are the flow plots in Fig 3A gated on? They show little to no CD4 T-cells in uncolonized mice, but there are many lymphocytes in the healthy gut so the plots look strange to me.

5. Did IL-17 levels change with Tmu colonisation? Is the boost in Th1 due to a reduction in IL-17/Th17 responses? That might explain a reduction in intestinal damage/inflammation as well as reduced PMN recruitment.

6. Lines 146-148 – authors indicate that IFN γ maintains intestinal integrity of the mice, but what evidence do they have that Tmu has any effect on integrity? Can the authors check barrier integrity and function in these mice, e.g. with FITC-dextran? There was no change in bacterial dissemination from the gut with or without Tmu, so I'm unclear where this focus on integrity comes from.

7. Lines 150-154: bit more explanation needed here on experimental set-up, was a bit difficult to follow as KEGG analysis often used in reference to RNAseq data. An introductory sentence to clarify that the data is based on metabolites would be helpful.

8. Were T-cells only sources of IFN γ affected by Tmu? What about ILCs? This analysis will definitely be important if authors explore the IL-17 side.

9. Arginase and iNOS staining of intestinal macrophages would be very informative to pinpoint cell types driving the metabolite changes and figure out how Tmu is doing that.

10. Histology of the colonisation and infection, with co-staining for relevant cell types (e.g. Tmu, Cdiff and PMN) is essential to understand how spatial relationships are affected by Tmu colonisation and get some indication as to how these phenotypes link together.

Reviewer #2 (Remarks to the Author):

This manuscript by Yang et al describes the metabolic and inflammatory interactions between the protozoan *Tritrichomonas musculus* and the human pathogen *Clostridioides difficile* in the mouse. The authors demonstrate that prior colonisation with *T. mu* reduces the severity of *C. difficile* infection and suggest a model involving changes in amino acid metabolism and influences on immune signalling and neutrophil recruitment.

This is a very interesting manuscript with the potential to add significantly to our understanding of how non-bacterial members of the microbiota might contribute to severity in CDI. However, I do have concerns about some of the data presented and also how this has been described and interpreted. These are described below, broadly in the order in which they occur in the main text.

Major issues

Figure 1

CFUs are reported for the spleen and mesenteric lymph nodes. This is a little surprising as this is not a common observation in *C. difficile* infected mice. However, I am a little concerned about the numbers reported. The spleen is approximately 80 mg and mesenteric lymph nodes are far smaller. The exact details of how the CFUs are determined are not entirely clear from the methods (plating 100 ul/ 10 ul Miles and Misra/something else?) so numbers reported here could be very close to the limit of detection. What is the limit of detection of your analysis method? This should be reported in the legend or directly on the graph.

Furthermore, colonisation as reported here does not differentiate spores and vegetative cells (this can have profound implications on disease), reports only a single time point (potentially missing important temporal differences) and, for caecal contents and faeces, relies solely on qPCR quantification which could conflate both viable and dead cells. Colonisation is commonly assessed by quantifying both total and spore CFUs at multiple time points. I assume quantification was done at 48 h but that isn't mentioned anywhere I can find.

The legend states n=4-10 but the body weight spreadsheet first tab includes data for 10 mice per test condition and all seem to have survived to the 48 h time point. Can the authors please clarify the numbers included in the analysis here and explain the discrepancy?

After the experiment reported in Figure 1 it appears that *C. difficile* in infected animals has not been quantified at all. This is extremely concerning given the wide variation that is typically seen in levels of colonisation, even in identical replicate experiments. The range of different interventions here could conceivably result in even more dramatic changes in colonisation but this is not assessed.

Figure 4 and Line 150 onwards

I was very confused by this as the text and figure legend don't describe what data the KEGG analysis has been done on. I eventually figured out that the authors had done metabolomics on caecal contents but this needs to be made much clearer. Line 154 refers to "two groups" but these aren't defined anywhere in the text or legend that I can find. I assume CDI and CDI/T.mu but I'm not certain. The heatmap is labelled "CD" and "T". Please add these details to the legend. The methods describe performing qPCR on mouse colons. I assume that's the origin of the data in panels H-L but that's not actually described in the text or legend and I was confused as to whether the described genes were from the mouse or T. mu.

Without knowing the level of *C. difficile* colonisation (and T. mu) in the mice in which caecal amino acid concentrations it's very hard to justify the assertion that the differences in concentrations are due to changes in host gene expression. As the authors point out both *C. difficile* and T. mu commonly utilise amino acids as C/N sources.

The authors also point out in the introduction that colonisation with T. mu is associated with changes in the composition of the microbiome. Could these changes not account for the observed differences in amino acid availability in the caecum?

Figure 5 and Line 168 onwards

Arginine was given via intraperitoneal injection (200 mg/kg for 7 days – once a day? Infused

over 7 days?). Does this definitely increase the concentration in the caecum? As the caecal concentration if not measured it's impossible to interpret this data. Similar concerns for the other amino acid supplements – does supplementation achieve a similar caecal concentration to that seen in the earlier experiments?

Line 171 – ASS enzyme – the mouse argininosuccinate synthase?

In general the figure legends are far too brief and are lacking key details needed to allow the reader to fully understand and interpret the presented data.

Minor issues

Clostridium difficile should be changed to *Clostridioides difficile* and this should be abbreviated to *C. difficile* consistently after first usage in-text.

Line 52: should call this fidaxomicin rather than fedamycin

Line 59: of the described non-antibiotic therapies only FMT has been demonstrated to be superior to standard antibiotic therapy.

What does NC refer to? In the methods this is described as “normal mice without *T. mu*” but that isn't defined in the text or figure legends.

Reviewer #3 (Remarks to the Author):

Yang and colleagues reports a murine commensal eukaryote, *Tritrichomonas musculus*, reduces severity of *Clostridioides difficile* infection in mice and explores mechanisms by which *T. mu* modulates host immunity. Authors demonstrate addition of *T. mu* to mice infected with *C. difficile* reduces an array of clinical metrics associated with infection without impacting bacterial burden or toxin levels in the gut. It is shown that the presence

of *T. mu* reduces pathogenic neutrophil infiltration associated with *C. Diff* infection along with pro-inflammatory cytokines in the intestinal epithelium and cecum. The authors then explore metabolic interactions between these two organisms and suggest that because *C. difficile* and *T. mu* both rely on amino acid metabolism. Through a series of experiments the authors show the role of arginine and ornithine in this system and use various diets and add-back experiments to demonstrate physiological relevance. Together, this work is interesting and add to a growing list of important ecological interactions in the gut ecosystem during *C. difficile* infection. However, several major concerns detailed below and a lack of contextualization with existing literature dampens enthusiasm.

The authors do not establish that the ornithine in the gut during *C. Diff* infection is derived from *C. difficile* and it is not clear why it was concluded that there is ornithine cross talk. Other explanations seem more likely based on previous studies. Ornithine from both host and other microbial sources has been shown to be a substrate for *C. difficile* (PMIDs: 36602960, 36385534, 34637781) during infection. Without evidence that *T. mu* is using ornithine derived from *C. difficile*, I don't agree that this is likely cross-feeding.

The authors describe a significant increase in fecal arginine, but Fig. 4E does not show that the addition of *T. mu* results in a statistical increase in fecal arginine. Furthermore, the authors demonstrate an up-regulation in both *Ass* and *iNos* expression, which would have competing effects on the arginine and citrulline pools. Additional explanation to clarify the relationship between fecal metabolites and gene expression seen in Fig. 4 is needed.

The rationale for exploring amino acid usage derives partially from the known utilization of arginine by *T. mu* for energy (L151). Arginine catabolism produces energy and the arginine dihydrolase pathway produces ornithine as a byproduct. However, the data in Fig. 4D-G suggests that *T. mu* is performing arginine biosynthesis in order to deplete ornithine and increase arginine. Additional experimentations and explanation is required to rationalize *T. mu* arginine biosynthesis rather than arginine catabolism.

Based on the increase in arginine seen in Fig. 4, authors hypothesize that supplementation of arginine would reduce the severity of infection. Arginine supplementation has been

shown to be protective during *C. difficile* infection in the previously noted studies. But more importantly, there's a discrepancy between the text and the methods here. The text describes mice being fed arginine while the method for this experiment describes IP injection of arginine. If the mice were fed arginine, how was it administered and what concentration? Additionally, the kinetics of disease in Fig. 5A-B suggest that the protection afforded by *T. mu* and arginine supplementation are not facilitated through the same mechanism. If the authors IP injected the mice, significant rationale is lacking to explain why IP injection would be physiologically relevant to arginine production by *T. mu* in the lumen. Seems as if IP infection would suggest a more systematic effect in this case.

The authors postulate that an ornithine-free diet would alleviate disease based on a depletion of fecal ornithine seen in Fig. 4G. This diet completely abrogates disease, more so than *T. mu* itself, suggesting either a different mechanism or additional mechanisms beyond those caused by *T. mu*. As stated above, since ornithine is known to play a significant role in *C. difficile* metabolism and fitness, it is possible that an ornithine-free diet is having a direct negative effect on *C. difficile*. Data on *C. difficile* burden and toxin production is needed to provide evidence to convince the reader of this.

Additional experimentation is needed to dissect the contributions of *T. mu* vs. arginine interaction with IFN γ . For example, performing *T. mu* colonization during *C. difficile* infection in a *Ifngr1*^{-/-} mouse line would provide more direct evidence of the proposed mechanism. Moreover, direct evidence of *T. mu* arginine biosynthesis would greatly add to Sections 4 and 5, since it is just not clear if the phenomena observed are direct or indirect.

Reviewer #1 (Remarks to the Author):

In this study, the authors examine how the protozoan parasite *Tmu* affected disease progression following infection with *C. diff*. They show that *Tmu* itself develops a symbiotic relationship in mice and doesn't affect the intestinal barrier or cause pathology. When mice are pre-colonised with *Tmu* before *Cdiff* infection, the authors find that *Tmu* in the gut alleviates overall disease scores without affecting *Cdiff* burden. They correlates this with reduced PMN recruitment, a boost in IFNG production and reduced arginase expression, which they theorise explains some of the metabolite changes they observe in the intestines.

This is an interesting paper that raises multiple questions. It is, however, largely descriptive in its findings, and there are many places where it isn't clear how phenotypes are linked or connected. It also isn't clear how *Tmu* is mediating any of the observed changes – there is no analysis on microbiome or experiments to address indirect effects. My specific comments and suggestions below.

1. My main comment for the authors is to address how *Tmu* is affecting these changes they observe. For example, what is the phenotype of germ-free mice when colonised with *T.mu* before *Cdiff* infection? Is it the same/similar effect? What happened to microbiome composition in *Tmu* colonised mice? Did it change – is that relevant? How do they know the metabolite changes they see were not caused by significant population changes occurring elsewhere? This would help us understand whether the effects of *T.mu* are direct on the host or indirectly mediated.

Response: Thank you for your suggestions.

1) To test whether the gut microbiota was changed upon *T.mu* colonization, we performed bacterial 16s rRNA gene (V3-V4 region) sequencing. The data are provided in the **Supplementary Fig. 2A-D**. Our results indicate that *T.mu* colonization indeed influences the microbial community that *Cd* might be encountered during its infection.

2) To investigate whether *T.mu* directly mediate protection against CDI, we designed two experiments. The first one is to use metronidazole (MNZ) to remove *T.mu* from *T.mu*-colonized mice. The second one is to colonize germ-free (GF) mice with *T.mu*. All of the experiments strongly suggest a direct role of *T.mu* in protection against CDI. Please check the **Supplementary Fig. 2E-H**, and the **main Fig. 2**.

3) To investigate whether *T.mu* regulation of arginine metabolism is related to gut microbiota, we measured the metabolite changes in GF mice after *T.mu* colonization. We found that *T.mu*-colonized GF mice still showed relatively lower ornithine levels and more putrescine, alanine levels in the cecal contents post *Cd* infection. Overall, these results suggest that *T.mu* alone can alter host arginine metabolism in GF mice infected with *C. difficile*. Please check the **Supplementary Fig. 6**.

2. What were the numbers of neutrophils in the lamina propria? Authors show frequency (which is striking), but that could reflect significant expansions of other cell types in the *T.mu* mice, without affecting neutrophil number. A more comprehensive analysis of the cellular composition of *Tmu* mice is needed.

Response: Thank you for your comment. We have repeated the murine CDI model and performed additional flow cytometry analysis, and enumerated the neutrophil numbers accordingly.

We found that *T.mu* colonization affects the neutrophil numbers in the lamina propria of CDI mice (Fig. 3A). In addition to neutrophils, we comprehensively analyzed Th1 cells (Fig. 4A), Th17 cells (Fig. 4B), ILCs (Supplementary Fig. 3A, 3B), macrophages (Supplementary Fig. 3C), $\gamma\delta$ T cells (Supplementary Fig. 3D) and Tc1 cells (Supplementary Fig. 3E). Our results showed that the *T.mu* colonization has no significant influence on the percentages and numbers of ILCs, macrophages, $\gamma\delta$ T cells, and Tc1 cells.

3. Related to above- the change in neutrophils in the gut is really striking. Was there any related changes to neutrophil production and circulation in the bone marrow? How are these effects on PMNs related to the arginine metabolism – which phenotype comes first?

Response: Thank you for your comment.

- 1) We analyzed the frequency and number of neutrophils in the blood and bone marrow accordingly.

We found that the numbers and percentages of neutrophils in the bone marrow were significantly dropped after CDI, and *T.mu* colonization reduced the magnitude of reduction (Fig. 3C). Moreover, we found that *T.mu* significantly reduced the expression of neutrophil chemokine CXCL1 (Fig. 3D-E). Together, the data suggests that *T.mu* inhibits the recruitment of neutrophils to the intestine during CDI.

- 2) In order to evaluate whether arginine metabolism is critical for *T.mu* to relieve CDI, the CDI mice were treated with arginine in the drinking water or fed with an ornithine-free diet.

Our data indicated that supplementation with 1% arginine in their drinking water or ornithine-free diet significantly reduced disease severity (Fig. 6C-G), decreased the recruitment of neutrophils in the intestine (Fig. 6H-I). Furthermore, α -MDLA (an inhibitor of ASS1) reversed the *T.mu*'s reducing effect on neutrophil recruitment in the colon post *Cd* infection (Fig. 7H). Together, these results suggest that arginine metabolism influences the neutrophil recruitment.

4. What are the flow plots in Fig 3A gated on? They show little to no CD4 T-cells in

uncolonized mice, but there are many lymphocytes in the healthy gut so the plots look strange to me.

Response: We apologize for the confusion generated by our previous data figure 3A. We have re-performed the flow cytometry analysis, and included updated data in the main Fig. 4A and Supplementary Fig. 3B. The CD4⁺ T cells are indeed presented in uninfected mice.

5. Did IL-17 levels change with *T.mu* colonisation? Is the boost in Th1 due to a reduction in IL-17/Th17 responses? That might explain a reduction in intestinal damage/inflammation as well as reduced PMN recruitment.

Response: Thank you for your suggestions. We analyzed the proportion and number of intestinal Th17 cells, as well as the expression of IL-17.

Our results showed that the proportion and numbers of intestinal Th17 cells were not significantly different between *T.mu*-colonized and un-colonized mice (Fig. 4B). The IL-17 protein levels were also the same (Fig. 4E).

6. Lines 146-148 – authors indicate that IFN γ maintains intestinal integrity of the mice, but what evidence do they have that *Tmu* has any effect on integrity? Can the authors check barrier integrity and function in these mice, e.g. with FITC-dextran? There was no change in bacterial dissemination from the gut with or without *Tmu*, so I'm unclear where this focus on integrity comes from.

Response: We apologize for the inaccurate description in the previous manuscript. The manuscript has been modified accordingly.

We checked barrier integrity and function with FITC-dextran in these mice, however, we could not draw any conclusion, as the measured values were very low. We also performed immunohistochemistry to measure the intestinal tight junction proteins ZO-1 and Occludin. The data are shown below. We could not find any differences between them (see Figure below).

7. Lines 150-154: bit more explanation needed here on experimental set-up, was a bit difficult to follow as KEGG analysis often used in reference to RNAseq data. An introductory sentence to clarify that the data is based on metabolites would be helpful.

Response: Thank you for your suggestions. We have changed our wording accordingly.

The following is our revised sentences:

“Next, we asked whether *T.mu* colonization can influence overall gut microbiome metabolomes. We therefore conducted untargeted metabolomic profiling of cecal content samples collected from *Cd*-infected mice with or without *T.mu* colonization. We found that many of the differentially enriched metabolites in the CDI group were amino acid metabolic intermediates (Fig. 5A), and within these metabolites, many of them were intermediates participating in arginine biosynthesis pathway (Fig. 5B). Compared with the CDI group, the relative levels of ornithine in the cecal contents were decreased, while the citrulline levels were instead increased in the CDI+*T.mu* group (Fig. 5C), suggesting a role of *T.mu* colonization in influencing certain amino acids richness (e.g., arginine/ornithine, etc.) in the gut lumen, a critical space shared by the gut microbial community.”

8. Were T-cells only sources of IFN γ affected by Tmu? What about ILCs? This analysis will definitely be important if authors explore the IL-17 side.

Response: Thank you for your suggestions.

We analyzed intestinal NK, $\gamma\delta$ T cells, Tc1 cells, ILCs and found that *T.mu* colonization did not affect the IFN- γ levels (Supplementary Fig. 3).

9. Arginase and iNOS staining of intestinal macrophages would be very informative to pinpoint cell types driving the metabolite changes and figure out how Tmu is doing that.

Response: Thank you for your suggestion. We did Arg1/[F4/80] and iNOS/[F4/80] co-immunostaining of colon tissues collected from *Cd*-infected WT mice accordingly.

Our data indicated that there was a reduction of Arg1⁺F4/80⁺ cells in the gut of *T.mu*-colonized mice post infection, while the iNos⁺F4/80⁺ cells were similar between the *T.mu*-colonized and *T.mu*-uncolonized mice (Supplementary Fig. 7). Notably, the expression of iNOS was strongly induced in the gut epithelium of *T.mu*-colonized mice (Supplementary Fig. 7). These results suggest that *T.mu* colonization influence not only alternatively activated M2 (ARG1⁺) macrophages, but also other type of cells that are involved in the regulation of intestinal arginine/ornithine metabolism during *Cd* infection.

10. Histology of the colonisation and infection, with co-staining for relevant cell types (e.g. Tmu, Cdiff and PMN) is essential to understand how spatial relationships are affected by Tmu colonisation and get some indication as to how these phenotypes link together.

Response: Thank you for your suggestions. We actively conducted this experiment.

Since there are no antibodies for *T.mu* and *C. difficile*, we designed and synthesized DNA probes for them and did FISH-staining. For neutrophils, we stained with anti-Ly6G antibody. Unfortunately, we have not been able to successfully stained all three of them together. We are still exploring different conditions and trying different probes. However, we hope that our newly generated data in the revised paper overall can support our conclusions.

Reviewer #2 (Remarks to the Author):

This manuscript by Yang et al describes the metabolic and inflammatory interactions between the protozoan *Trichomonas musculus* and the human pathogen *Clostridioides difficile* in the mouse. The authors demonstrate that prior colonisation with *T. mu* reduces the severity of *C. difficile* infection and suggest a model involving changes in amino acid metabolism and influences on immune signalling and neutrophil recruitment.

This is a very interesting manuscript with the potential to add significantly to our understanding of how non-bacterial members of the microbiota might contribute to severity in CDI. However, I do have concerns about some of the data presented and also how this has been described and interpreted. These are described below, broadly in the order in which they occur in the main text.

Major issues

Figure 1

CFUs are reported for the spleen and mesenteric lymph nodes. This is a little surprising as this is not a common observation in *C. difficile* infected mice. However, I am a little concerned about the numbers reported. The spleen is approximately 80 mg and mesenteric lymph nodes are far smaller. The exact details of how the CFUs are determined are not entirely clear from the methods (plating 100 ul/ 10 ul Miles and Misra/something else?) so numbers reported here could be very close to the limit of detection. What is the limit of detection of your analysis method? This should be reported in the legend or directly on the graph.

Response: Sorry for the misleading description in the previous manuscript. We agree with the reviewer's opinion. The CFU reported for the spleen and MLN is too low to be trusted, and we therefore removed these data. We only included the CFU in the faeces and cecal contents data in the revised manuscript.

Furthermore, colonisation as reported here does not differentiate spores and vegetative cells (this can have profound implications on disease), reports only a single time point (potentially missing important temporal differences) and, for caecal contents and faeces, relies solely on qPCR quantification which could conflate both viable and dead cells. Colonisation is commonly assessed by quantifying both total and spore CFUs at multiple time points. I assume quantification was done at 48 h but that isn't

mentioned anywhere I can find.

Response: Thank you for your suggestions.

We assessed *C. difficile* colonization by quantifying both vegetative and spore CFUs in the fecal (at 12h, 24h), cecal contents (at 48h) and cecum (at 48h) after infection, and we also tested toxin TcdB titers at the same time. Please check the data in the main Fig. 1I-M.

The legend states n=4-10 but the body weight spreadsheet first tab includes data for 10 mice per test condition and all seem to have survived to the 48 h time point. Can the authors please clarify the numbers included in the analysis here and explain the discrepancy?

Response: We have repeated the mouse model for 2 times, the n=4 is only for the CFU count experiment.

Now, we have done the third experiments, and assessed *C. difficile* colonization by quantifying both vegetative and spores CFUs, and toxin levels after infection *C. difficile* were also measured (now, the n=8 for this new experiment). Please check Fig 1I-M.

After the experiment reported in Figure 1 it appears that *C. difficile* in infected animals has not been quantified at all. This is extremely concerning given the wide variation that is typically seen in levels of colonisation, even in identical replicate experiments. The range of different interventions here could conceivably result in even more dramatic changes in colonisation but this is not assessed.

Response: Thank you for your suggestions.

We have now added vegetative biomass, spores biomass and toxin titers in every CDI experiments that we have done. Please check Fig 1I-M.

Figure 4 and Line 150 onwards

I was very confused by this as the text and figure legend don't describe what data the KEGG analysis has been done on. I eventually figured out that the authors had done metabolomics on caecal contents but this needs to be made much clearer. Line 154 refers to "two groups" but these aren't defined anywhere in the text or legend that I can find. I assume CDI and CDI/T.mu but I'm not certain. The heatmap is labelled "CD" and "T". Please add these details to the legend. The methods describe performing qPCR on mouse colons. I assume that's the origin of the data in panels H-L but that's not actually described in the text or legend and I was confused as to whether the described genes were from the mouse or T. mu.

Response: We apologize for the confusion caused by our ambiguous description.

The following is our revised sentences:

“Next, we asked whether *T.mu* colonization can influence overall gut microbiome metabolomes. We therefore conducted untargeted metabolomic profiling of cecal content samples collected from *Cd*-infected mice with or without *T.mu* colonization. We found that many of the differentially enriched metabolites in the CDI group were amino acid metabolic intermediates (Fig. 5A), and within these metabolites, many of them were intermediates participating in arginine biosynthesis pathway (Fig. 5B). Compared with the CDI group, the relative levels of ornithine in the cecal contents were decreased, while the citrulline levels were instead increased in the CDI+*T.mu* group (Fig. 5C), suggesting a role of *T.mu* colonization in influencing certain amino acids richness (e.g., arginine/ornithine, etc.) in the gut lumen, a critical space shared by the gut microbial community.”

Without knowing the level of *C. difficile* colonisation (and *T. mu*) in the mice in which caecal amino acid concentrations it's very hard to justify the assertion that the differences in concentrations are due to changes in host gene expression. As the authors point out both *C. difficile* and *T. mu* commonly utilise amino acids as C/N sources.

Response: Thank you for your suggestions.

We have now added the information in the **main Fig. 1I-M, and the Supplementary Fig. 1D**.

The authors also point out in the introduction that colonisation with *T. mu* is associated with changes in the composition of the microbiome. Could these changes not account for the observed differences in amino acid availability in the caecum?

Response: Thank you for your comments.

- 1) To investigate whether the gut microbiota was changed upon *T.mu* colonization after *Cd* infection, we performed 16s rRNA gene sequencing in mice cecal contents samples. Our results indicated that *T.mu* indeed change the intestinal microbial landscape (**Supplementary Fig. 2A-D**).
- 2) We agree with the reviewer's comment, the overall differences in amino acid availability in the cecum are the net results from *T.mu*, *Cd*, other microbes, and the host. However, we speculate that the ability of *T.mu* to metabolize arginine and ornithine may be important for its protective role in *Cd* infection. The reason is related to the fact that the ornithine is important nutrients utilized by *Cd* for its colonization. Ornithine oxidative metabolism in *Cd* supports its non-inflammatory asymptomatic colonization in mice. In addition, the modulation of the availability of arginine and ornithine levels by *T.mu* are likely to be sensed by both the host and the gut microbial community, which can, in return, influence the *Cd* functions.

Figure 5 and Line 168 onwards

Arginine was given via intraperitoneal injection (200 mg/kg for 7 days – once a day? Infused over 7 days?). Does this definitely increase the concentration in the caecum? As the caecal concentration if not measured it's impossible to interpret this data. Similar concerns for the other amino acid supplements – does supplementation achieve a similar caecal concentration to that seen in the earlier experiments?

Response: We agree with the reviewer's comment. IP injection is not physiologically relevant to gut metabolic changes.

We added new experiments in our revised manuscript. We added 1% arginine in drinking water to feed the mice, we also used an ornithine-free diet to feed the mice. The gut luminal concentrations of arginine and ornithine were monitored accordingly.

Line 171 – ASS enzyme – the mouse argininosuccinate synthase?

In general the figure legends are far too brief and are lacking key details needed to allow the reader to fully understand and interpret the presented data.

Response: Thank you for your comments. We have provided the information as needed.

Minor issues

Clostridium difficile should be changed to *Clostridioides difficile* and this should be abbreviated to *C. difficile* consistently after first usage in-text.

Response: Thank you for your suggestions. We have changed *Clostridium difficile* to *Clostridioides difficile*.

Line 52: should call this fidaxomicin rather than fedamycin

Response: Thank you for your suggestions. We have changed fedamycin to fidaxomicin.

Line 59: of the described non-antibiotic therapies only FMT has been demonstrated to be superior to standard antibiotic therapy.

Response: Thank you for your suggestions. We have now made a revision.

What does NC refer to? In the methods this is described as “normal mice without T. mu” but that isn't defined in the text or figure legends.

Response: Thank you for your suggestions. We have added to the information accordingly.

Reviewer #3 (Remarks to the Author):

Yang and colleagues reports a murine commensal eukaryote, *Tritrichomonas musculus*, reduces severity of *Clostridioides difficile* infection in mice and explores mechanisms by which *T. mu* modulates host immunity. Authors demonstrate addition of *T. mu* to mice infected with *C. difficile* reduces an array of clinical metrics associated with infection without impacting bacterial burden or toxin levels in the gut. It is shown that the presence of *T. mu* reduces pathogenic neutrophil infiltration associated with *C. Diff* infection along with pro-inflammatory cytokines in the intestinal epithelium and cecum. The authors then explore metabolic interactions between these two organisms and suggest that because *C. difficile* and *T. mu* both rely on amino acid metabolism. Through a series of experiments the authors show the role of arginine and ornithine in this system and use various diets and add-back experiments to demonstrate physiological relevance. Together, this work is interesting and add to a growing list of important ecological interactions in the gut ecosystem during *C. difficile* infection. However, several major concerns detailed below and a lack of contextualization with existing literature dampens enthusiasm.

The authors do not establish that the ornithine in the gut during *C. Diff* infection is derived from *C. difficile* and it is not clear why it was concluded that there is ornithine cross talk. Other explanations seem more likely based on previous studies. Ornithine from both host and other microbial sources has been shown to be a substrate for *C. difficile* (PMIDs: 36602960, 36385534, 34637781) during infection. Without evidence that *T. mu* is using ornithine derived from *C. difficile*, I don't agree that this is likely cross-feeding.

Response:

We agree with the reviewer's comment.

What we really want to say is that the presence of *T.mu* influences not only the gut microbiota amino acids profiles, but also the host intestinal metabolic activities, especially for the cross-kingdom inter-related arginine-ornithine metabolic axis.

The rationale for exploring amino acid usage derives partially from the known utilization of arginine by *T. mu* for energy (L151). Arginine catabolism produces energy and the arginine dihydrolase pathway produces ornithine as a byproduct. However, the data in Fig. 4D-G suggests that *T. mu* is performing arginine biosynthesis in order to deplete ornithine and increase arginine. Additional experimentations and explanation is required to rationalize *T. mu* arginine biosynthesis rather than arginine catabolism.

Response:

We have not described our data very clearly in the previous manuscript. What the bioinformatic data really means is that the mice that are infected with CD have many metabolic intermediates involved in the arginine biosynthesis pathway. It doesn't mean that the arginine biosynthesis pathway per se is upregulated or not in the CDI mice.

Based on the increase in arginine seen in Fig. 4, authors hypothesize that supplementation of arginine would reduce the severity of infection. Arginine supplementation has been shown to be protective during *C. difficile* infection in the previously noted studies. But more importantly, there's a discrepancy between the text and the methods here. The text describes mice being fed arginine while the method for this experiment describes IP injection of arginine. If the mice were fed arginine, how was it administered and what concentration? Additionally, the kinetics of disease in Fig. 5A-B suggest that the protection afforded by *T. mu* and arginine supplementation are not facilitated through the same mechanism. If the authors IP injected the mice, significant rationale is lacking to explain why IP injection would be physiologically relevant to arginine production by *T. mu* in the lumen. Seems as if IP infection would suggest a more systematic effect in this case.

Response: We agree with the reviewer's comments. Now, we modified the model by adding arginine in drinking water.

When mice treated with 1% arginine in their drinking water, this increased arginine in the gut lumen (Fig. 6A). We also used an ornithine-free diet to fed the mice, this decreased ornithine in the gut lumen (Fig. 6B).

Supplementation with arginine or ornithine-free diet significantly reduced parameters of disease severity (Fig. 6C-G). Further, arginine and ornithine-free diet decreased the recruitment of neutrophils, the secretion of the chemokine CXCL1 and the expression of IL-1 β in the intestinal of CDI mice (Fig. 6H-K). Supplementation with arginine significantly reduced toxin titer (Fig. 6N).

The authors postulate that an ornithine-free diet would alleviate disease based on a depletion of fecal ornithine seen in Fig. 4G. This diet completely abrogates disease, more so than *T. mu* itself, suggesting either a different mechanism or additional mechanisms beyond those caused by *T. mu*. As stated above, since ornithine is known to play a significant role in *C. difficile* metabolism and fitness, it is possible that an ornithine-free diet is having a direct negative effect on *C. difficile*. Data on *C. difficile* burden and toxin production is needed to provide evidence to convince the reader of this.

Response: We agree with the reviewer's comments.

When treated CDI mice with ornithine-free diet, we found that the concentration of ornithine in the gut lumen was lower than that in the intestinal lumen of mice co-infected with *T.mu* (Fig. 6B). We showed that *T.mu* can reduce the virulence of *C. difficile* without affecting its colonization. Ornithine-free diet intervention can alleviate *C. difficile* virulence and colonization. In addition to reducing ornithine in CDI mice, *T.mu* co-infection also increased the gut lumen arginine, putrescine, alanine and 5-aminovalerate (Fig. 5F, Fig. 5G, Supplementary Fig. 6B). It has been reported that arginine can reduce the virulence of *C. difficile*. Putrescine has been reported to accelerate intestinal epithelial renewal and regulate host intestinal homeostasis. Alanine

is produced by *C. difficile* through ornithine oxidation metabolism. *C. difficile* utilizes oxidative ornithine metabolism to support its asymptomatic colonization. We speculate that co-infection of *C. difficile* with T.mu, on one hand, *C. difficile* utilizes ornithine oxidation metabolism to promote its low-virulence and low-inflammatory colonization, while on the other hand, arginine also weakens the virulence of *C. difficile*. Therefore, the protective effect of *T.mu* against *C. difficile* and the ornithine deficiency effect are not completely the same.

Additional experimentation is needed to dissect the contributions of T. mu vs. arginine interaction with IFN γ . For example, performing T. mu colonization during *C. difficile* infection in a *Ifngr1*^{-/-} mouse line would provide more direct evidence of the proposed mechanism. Moreover, direct evidence of T. mu arginine biosynthesis would greatly add to Sections 4 and 5, since it is just not clear if the phenomena observed are direct or indirect.

Response: We agree with the reviewer's comment.

- 1) We modeled T.mu-*C. difficile* co-colonization in *Ifngr*^{-/-} mice to explore the role of IFN- γ in *T.mu* protection against CDI.

Our results showed that T.mu did not relieve clinical symptoms caused by *C. difficile* in *Ifngr*^{-/-} mice (Fig. 4F, 4G). Similar to the CDI group, *Ifngr*^{-/-} mice co-infected with T.mu showed more serious colonic atrophy, a large number of inflammatory cells infiltration in the cecum, lower colon length (Fig. 4H-J) and a significant decrease in goblet cells and MUC2 protein (Fig. 4K, 4L). Moreover, administration of recombinant IFN- γ protein to CDI mice reduces susceptibility to *C. difficile*, mimicking the effect of T.mu intervention.

- 2) We blocked the host Ass1 enzyme by using a small molecular inhibitor.

We found blocking the host arginine metabolic pathway can abolish the protective effect of T.mu on CDI. Please check Fig 7.

REVIEWER COMMENTS

Reviewer #1 (Remarks to the Author):

In the revised version, the authors have added numerous extra experiments, the most important of which are the germ-free experiments which strengthen some of their original conclusions and are an important addition. I would stress to the authors the importance of highlighting changes in revised manuscripts - reviewing a revised manuscript is much harder when you have to search for the changes. Despite that, I think the authors have done a good job of answering my original queries. I think some additional improvements could still be made to figure presentation (e.g. labelling which graphs are germ-free mice and so on because this was not immediately obvious).

Reviewer #2 (Remarks to the Author):

This manuscript is a resubmission of a paper that I reviewed in June of last year. When I first saw this manuscript I had concerns about aspects of the experimental design and the presentation, description and interpretation of some of the data.. The authors have done an admirable job of addressing these concerns and I believe the current version is significantly improved. Overall, I can identify no significant issues in this version and I think this version will be of great interest in the field. I did note a few minor textual issues, described below, that you may want to address in a final version.

Minor comments:

I dislike non-standard abbreviations of species names (e.g. Cd) but I will leave this to Nature editorial policy!

Line 66: referring to the spore as the main pathogen is an inaccurate representation. I assume you mean that it's the infectious agent?

Line 70: I'd probably rephrase this as there is little evidence for evolved resistance to antibiotics in *C. difficile* in contrast to other well-studied AMR pathogens. Aside from sensitivity to fluoroquinolones (and possibly metronidazole in a minority or lineages) the

AMR profile of *C. difficile* hasn't changed much in the last couple of decades of focused study.

Line 114: "that are molecular genetically identical to" – that is genetically identical to...

Line 138: TcdB not Tcd-B (and line 171 again).

Line 157: that Cd might encounter during.

Reviewer #3 (Remarks to the Author):

In this revised manuscript, the authors have strengthened their manuscript by performing significant new experimentation. Specifically, authors have improved the depth of the study by including germ-free mouse infections, an *lfngr*^{-/-} mouse model, improved dietary studies, a small-molecule inhibitor of the *Ass1* enzyme, and microbiome studies. The authors have addressed a number of major points of concern by myself and other Reviewers, and specifically clarified discrepancy between the arginine diet study data and methods. Overall, the manuscript is much improved, but below are several important remaining concerns.

The authors added some references, but there is still a lack of citation of important literature in this space. Similar studies exploring the arginine-ornithine axis, microbe-microbe interactions, and arginine supplementation during *C. difficile* infection have not been highlighted (e.g. PMID: 36385534). This is particularly important, as the authors show the arginine and ornithine manipulation in the diet alters disease outcomes in their model of infection. It is suggested by the authors in their overall model that this is driven by alteration to the host; however, the data shows that dietary manipulation impacts *C. difficile* toxin levels. This has been shown previously and is likely an effect of ornithine and arginine on *C. difficile* virulence. This effect should be expanded on by the authors as a potentially equally as important role of the arginine-ornithine metabolic axis on disease outcome and previous literature cited.

The microbiome figures need clarification, and the figure legends need significantly more

detail. In the methods, it is noted that the ordination (Supplementary Fig 2a) is principal coordinate analysis (PCoA), but the axes of the ordination suggest it is principal component (PC). This should be fixed and the axes should read PCoA 1 and PCoA 2. Additionally, it is not clear what the legend for the heatmap represents in Supplementary Fig 2C. Is this relative abundance? Correlation analysis?

It is recommended that the authors use a color-blind friendly palette for their figures.

Reviewer #1 (Remarks to the Author):

In the revised version, the authors have added numerous extra experiments, the most important of which are the germ-free experiments which strengthen some of their original conclusions and are an important addition. I would stress to the authors the importance of highlighting changes in revised manuscripts - reviewing a revised manuscript is much harder when you have to search for the changes. Despite that, I think the authors have done a good job of answering my original queries. I think some additional improvements could still be made to figure presentation (e.g. labelling which graphs are germ-free mice and so on because this was not immediately obvious).

Response:

Thank you for your suggestions.

- 1) To highlight the changes in the revised manuscript, we will send two versions of the manuscript, one of which uses tracking system.
- 2) For figure presentation: we added more details accordingly.

Reviewer #2 (Remarks to the Author):

This manuscript is a resubmission of a paper that I reviewed in June of last year. When I first saw this manuscript, I had concerns about aspects of the experimental design and the presentation, description and interpretation of some of the data. The authors have done an admirable job of addressing these concerns and I believe the current version is significantly improved. Overall, I can identify no significant issues in this version and I think this version will be of great interest in the field. I did note a few minor textual issues, described below, that you may want to address in a final version.

Minor comments:

1. I dislike non-standard abbreviations of species names (e.g. Cd) but I will leave this to Nature editorial policy!

Response:

Thank you for your suggestions. Considering that Cd is not a standard abbreviation for *Clostridioides difficile*, Cd is changed back to *C. difficile* in the revised manuscript.

2. Line 66: referring to the spore as the main pathogen is an inaccurate representation. I assume you mean that it's the infectious agent?

Response:

Thank you, we have changed the sentence as your suggestion.

3. Line 70: I'd probably rephrase this as there is little evidence for evolved resistance to antibiotics in *C. difficile* in contrast to other well-studied AMR pathogens. Aside from sensitivity to fluoroquinolones (and possibly metronidazole in a minority or lineages) the AMR profile of *C. difficile* hasn't changed much in the last couple of decades of focused study.

Response:

We have rephrased the sentence accordingly.

The revised sentence is as the following:

“Antibiotics are the main therapy for CDI. However, the acquisition of resistance to commonly used antibiotics facilitates the epidemic spread of hypervirulent *C. difficile* strains”.

Pls check Lines 66-68 of the revised manuscript.

4. Line 114: “that are molecular genetically identical to” – that is genetically identical to...

Response:

We have changed "that are molecular genetically identical to" to "that is genetically identical to". Pls check Line 112 of the revised manuscript.

5. Line 138: TcdB not Tcd-B (and line 171 again).

Response:

We have changed the "Tcd-B" to "TcdB" accordingly.

6. Line 157: that Cd might encounter during.

Response:

We have changed the " be encountered " to "encounter".

Reviewer #3 (Remarks to the Author):

In this revised manuscript, the authors have strengthened their manuscript by performing significant new experimentation. Specifically, authors have improved the depth of the study by including germ-free mouse infections, an *Ifngr*^{-/-} mouse model,

improved dietary studies, a small-molecule inhibitor of the Ass1 enzyme, and microbiome studies. The authors have addressed a number of major points of concern by myself and other Reviewers, and specifically clarified discrepancy between the arginine diet study data and methods. Overall, the manuscript is much improved, but below are several important remaining concerns.

1. The authors added some references, but there is still a lack of citation of important literature in this space. Similar studies exploring the arginine-ornithine axis, microbe-microbe interactions, and arginine supplementation during *C. difficile* infection have not been highlighted (e.g. PMID: 36385534). This is particularly important, as the authors show the arginine and ornithine manipulation in the diet alters disease outcomes in their model of infection. It is suggested by the authors in their overall model that this is driven by alteration to the host; however, the data shows that dietary manipulation impacts *C. difficile* toxin levels. This has been shown previously and is likely an effect of ornithine and arginine on *C. difficile* virulence. This effect should be expanded on by the authors as a potentially equally as important role of the arginine-ornithine metabolic axis on disease outcome and previous literature cited.

Response:

We agree with the reviewer's opinion.

The reference (PMID: 36385534) has been added accordingly. Pls check Lines 370-372.

Similar studies exploring the arginine-ornithine axis, microbe-microbe interactions, and arginine supplementation during *C. difficile* infection have been highlighted in the discussion. Pls check Lines 358-370.

We also briefly discussed the results of controlling dietary arginine and ornithine to alter disease, highlighting the dual role of the arginine-ornithine metabolic axis on host immunity and *C. difficile* virulence coordinately regulates *C. difficile* infection outcome. Pls check Lines 421-425.

2. The microbiome figures need clarification, and the figure legends need significantly more detail. In the methods, it is noted that the ordination (Supplementary Fig 2a) is principal coordinate analysis (PCoA), but the axes of the ordination suggest it is principal component (PC). This should be fixed and the axes should read PCoA 1 and PCoA 2. Additionally, it is not clear what the legend for the heatmap represents in Supplementary Fig 2C. Is this relative abundance? Correlation analysis?

Response:

Thank you for your suggestions.

1) We revised the axis labels for Supplementary Figure 2A accordingly.

2) For the heatmap of the Supplementary Figure 2C:

In the original version, we used the lg value of the absolute counts for heatmap presentation. For better comparison, we changed the heatmap plot in the revised manuscript by using the relative taxa abundance for heatmap color scale presentation.

3) More descriptive details were added in the figure legend accordingly.

3. It is recommended that the authors use a color-blind friendly palette for their figures.

Response:

Thank you for your suggestions. We have changed the color scheme accordingly to be color-blind friendly.

REVIEWERS' COMMENTS

Reviewer #3 (Remarks to the Author):

The authors have addressed all of my remaining concerns in their latest revision.